# Evaluation of snow cover and snow depth on the Qinghai-Tibetan Plateau derived from passive microwave remote sensing

Liyun Dai[1,4], Tao Che [1,2], Yongjian Ding [3], Xiaohua Hao [1]

[1]Key Laboratory of Remote Sensing of Gansu Province, Heihe Remote Sensing Experimental Research Station, Cold and Arid Regions Environmental and Engineering Research Institute, Chinese Academy of Sciences, Lanzhou 730000, China

[2]Center for Excellence in Tibetan Plateau Earth Sciences, Chinese Academy of Sciences, Beijing 100101, China

[3] State Key Laboratory of Cryospheric Sciences, Cold and Arid Regions Environmental and Engineering Research Institute, Chinese Academy of Sciences, Lanzhou 730000, China

[4]Jiangsu Center ofr Collaborative Innovation in Geographical Information Resource Development and Application, Nanjing 21003, China

*Correspondence to*: Tao Che (chetao@lzb.ac.cn)

**Abstract.** Snow cover on the Qinghai-Tibetan plateau (QTP) plays a significant role in the global climate system and is an important water resource for rivers in the high elevation region of Asia. At present, passive microwave (PMW) remote sensing data are the only efficient way to monitor temporal and spatial variations in snow depth at large scale. However, existing snow depth products show the largest uncertainties across the QTP. In this study, MODIS fractional snow cover product, point, line and intense sampling data  are synthesized to evaluate the accuracy of snow cover and snow depth derived from PMW remote sensing data and to analyze the possible causes of uncertainties. The results show that the accuracy of snow cover extents varies spatially and depends on the fraction of snow cover. Based on the assumption that grids with MODIS snow cover fraction > 10 % are regarded as snow cover, the overall accuracy in snow cover is 66.7 %, overestimation error is 56.1 %, underestimation error is 21.1 %, commission error is 27.6 % and omission error is 47.4 %. The commission and overestimation errors of snow cover primarily occur in the northwest and southeast areas with low ground temperature. Omission error primarily occurs in cold desert areas with shallow snow, and underestimation error mainly occurs in glacier and lake areas. With the increase of snow cover fraction, the overestimation error decreases and the omission error increases. Comparison between snow depths measured in field experiments, measured at meteorological stations and estimated across the QTP shows that agreement between observation and retrieval improves with an increasing number of observation points in a PMW grid. The misclassification and errors between observed and retrieved snow depth are associated with the relatively coarse resolution of PMW remote sensing, ground temperature, snow characteristics and topography. To accurately understand the variation in snow depth across the QTP, new algorithms should be developed to retrieve snow depth with higher spatial resolution and should consider the variation in brightness temperatures at different frequencies emitted from ground with changing ground features.

## 1 Introduction

The Qinghai-Tibetan plateau (QTP) is considered the third pole of the world and the Asian water tower (Kang et al., 2010; Wu et al., 2003; Wang et al., 2015; Immerzeel et al., 2010; Xu et al., 2008). Snow cover over it plays a significant role in the climate change and hydrological circle. Due to its importance regionally and globally and evident change (Shi and Wang, 2015), more attention should be paid to the snow cover variability across the QTP. Monitoring snow cover variability requires reliable snow depth and snow cover data.

Traditional station observation is used to monitor inter-annual variation of snow depth at local or regional scales. Inter-annual changes in snow cover and depth in Russia was analyzed using snow depths observed at 856 stations (Bulygina et al., 2009). Zhong et al. (2014) used station observation to analyze the snow density of Eurasian region. It was also use to long time series analysis due to its long history (Gafurov et al., 2015). However, meteorological station data do not always represent the snow status of a region, especially in regions with few stations, such as the QTP, although there are some studies that have reported spatiotemporal variation across the QTP using an interpolation method based on meteorological stations (Wang et al., 2009; You et al., 2011). In the absence of a large, distributed network of meteorological stations, remote sensing becomes a necessary technique.

Optical remote sensing can be used to identify snow cover extent accurately using the normalized difference of snow index (NDSI) method due to its high reflectance in the optical band and low reflectance in the near infrared band (Hall et al., 2002 and 2007). However, the drawback of optical remote sensing is that clouds mask snow data on most the days during the snow season. Therefore, 8-day and 16-day composite snow cover products are produced to eliminate cloud cover (Hall et al., 2002 and 2007). Daily cloud-free snow cover products were also produced using temporal or spatial interpolation algorithms (Tang et al., 2013; Hall et al., 2010; Gafurov and Bardossy, 2009; Parajka et al., 2010). However, for the strong spatial heterogeneity and rapid snow cover changes across the QTP, interpolation algorithms do not work under conditions of continuous multi-day cloud cover or for large areas. Therefore, in the cloud-covered areas, snow cover derived from passive microwave (PMW) remote sensing, which is independent of sunlight, has been used to supplement optical remote sensing (Liang et al., 2008; Gao et al., 2012; Deng et al., 2015). The data from the combination of these two techniques provides information masked by clouds and improves the temporal resolution of snow cover products. Many combined snow cover

products have been used in climate change and hydrological analysis (Barnett et al., 2005; Wang et al., 2015; Brown and Robinson, 2011; Choi et al., 2010 ). However, the accuracy of snow cover from PMW directly influences the accuracy of the combined snow cover product. In addition, although optical remote sensing is an efficient way to monitor spatial snow cover with high resolution, it cannot penetrate snowpack and obtain snow depth.

PMW is the only efficient way to monitor the spatial and temporal variation of snow depth. It is used to identify snow cover based on the volume scattering of snow particles. Brightness temperature emitted from the ground goes through snowpack and is scattered by snow particles. Furthermore, the scatter intensity at low frequency is weaker than that at high frequency, and the difference increases with number of snow particles. Therefore, regional and local snow depths have been retrieved based on the microwave spectral gradient method (Kelly et al., 2003; Pullianen et al., 2006; Dai et al., 2012; Jiang et al.,

2014), and these snow depth products have been widely used in climate change and vegetation variation, frozen soil detection, and hydrological cycle studies (Gao et al., 2012; Yu et al., 2013; Xu et al., 2009).

However, there are uncertainties with these snow depth products, and some assessments had been performed on them. The NASA snow water equivalent product derived from Advanced Microwave Scanning Radiometer for Earth Observing System (AMSR-E) generally tends to underestimate snow depth in North America (Tedesco and Narvekar, 2010) when compared

with World Meteorological Organization (WMO) and Snow Data Assimilation System (SNODAS), but overestimate in Northwest and Northeast of China (Dai et al., 2012; Che et al., 2016) when compared with Meteorological station and field work observations. These authors pointed out that the errors primarily came from the spatio-temporal variability of grain size and forest cover. It was because of the strong influence of grain size that mass investigation of snow characteristics were performed to obtain the a priori information of snow characteristics in Northwest and northeast China to improve the

simulation of brightness temperature and retrieval accuracy of snow depth. And some research assimilated snow depth observed at stations, or built a local empirical relationship between snow depth observations and spectral gradients to improve the snow depth retrieval accuracy in some regions (Dai et al., 2012; Che et al., 2016, 2008; Pullianen, 2006). In order to reduce the influence of forest, forest transimissivities at different frequencies were absorbed in the algorithm, or the special equations were built between snow depth and TBD at different types of forest, to estimate snow depth

more accurately at forest areas (Che et al., 2016, Pullianen, 2006).  However, uncertainties still exist for the snow depth

over QTP except grain size and forest. Based on existing research, the snow cover over the QTP was also overestimated compared with the optical snow extent products (Frei et al., 2012; Armstrong and Brodzik, 2002). Across the QTP, meteorological stations are rare and primarily distributed in the valley with low elevation. Snow depth observed at these stations does not represent the snow status of the grid they are located on, and so it is unclear if data assimilation and an empirical equation will work to improve snow depth accuracy. It has also been reported that snow cover across the QTP is overestimated by PMW algorithms compared to IMS snow cover products, which was caused by the large elevation with a thinner atmosphere (Savoie et al., 2009). Compared with meteorological station observation, AMSR-E SWE also presented overestimation (Yang et al., 2015). But, Smith and Bookhagen (2016) thought Tibet lacks an extensive and reliable ground-weather station network, therefore, they did not rely on in-situ data, but focused on the factors reducing the reliability of SWE estimates from satellite-based PMW data by comparing different satellite sensors. They found that satellite look angle and elevation showed very low influence on SWE variability. Thus, it seemed that general overestimation was undoubted for the QTP, but at present, there is no definitive answer to what are the causes and where do the overestimates occur over the QTP.

Therefore, the purposes of this study are to provide a reliable evaluation or assessment of the ability of passive microwave to detect snow cover and snow depth across the QTP using MODIS snow cover product and in situ observation data, analyze the cause of uncertainties, and provide reference for the use of PMW snow depth data and improvements to the retrieval algorithm for snow depth across the QTP.

## 2. DATA

### 2.1 MODIS snow cover fraction

The Terra/Aqua MODIS Level 3, 500m daily fractional snow cover products (MOD10A1 and MYD10A1) were obtained from the National Snow and Ice Data Center (NSIDC) from 1 January, 2003 to 31 December, 2014 (Hall et al., 2006). These products derived from MODIS were generated based on the regression relationship between normalized difference snow index (NDSI) and snow cover fraction (SCF). The relationship equation is SCF = a+ b* NDSI, and the coefficients "a" and "b" and NDSI vary with sensors. Coefficients "a" and "b" are -0.01 and 1.45 for MODA1, and -0.64 and 1.91 for MYDA, respectively. NDSI is the function of band 4 and band 6 for Terra (MODA1), and of band 4 and band 7 for

Aqua (MYDA1) (Salomonson and Appel, 2006). To develop a relationship between NDSI and SCF within a MODIS 500-m pixel, it was necessary to utilize a source of ground truth. In this algorithm, several Landsat scenes covering a wide variety of snow-cover conditions were selected, and every 30-m pixel of Landsat scene was classified as snow or no-snow. The number of snow-cover pixel for Landsat in a MODIS grid and the total number of Landsat pixels in a MODIS grid were calculated.

The ratio of them was the ground truth of SCF. When the derived snow cover fractions were compared to Landsat-7 Enhanced Thematic Mapper ground-truth observations covering a substantial range of snow cover conditions, the correlation coefficients were near 0.9 and the RMSE were near 0.10 (Salomonson and Appel, 2006).

## 2.2 Passive microwave brightness temperature and snow depth product

The AMSR-E, which measures twelve bands of six frequencies, was operated from the NASA EOS Aqua Satellite and

provided global passive microwave measurements of the earth from June, 2002 to October, 2010. To provide consistency of different frequencies with different footprints, the brightness temperature was resampled to an equal-area scalable earth grid (EASE-Grid) with a resolution of 25 km, approximately equal to $0.1°$ across the QTP. In this study, the brightness temperature at 18.7 GHz, 36.5 GHz at both vertical and horizontal polarization ($TB_{18H}$, $TB_{18V}$, $TB_{36H}$, $TB_{36V}$), 23.8 GHz, and 89.0 GHz at vertical polarization ($TB_{23V}$, $TB_{89V}$) from 1 January, 2003, to 31 December, 2008, were used to identify

snow cover and derive snow depth across the QTP.

The Advanced Microwave Scanning Radiometer-2 (AMSR2) carried on the Global Change Observation Mission (GCOM) was launched on May 18, 2012 (Imaoka et al. 2010), and provided brightness temperature from July 3, 2012. The AMSR2 sensor was the continuation of AMSR-E and has the same channels as the AMSR-E but a slightly smaller footprint. The AMSR2 brightness temperatures from November, 2013 to March, 2014 were used to derive the snow depth in the field

experiment areas. In order to avoid the influence of liquid water, the descending orbit data was used and when the brightness temperature at 36GHz for vertical polarization was more than 265 K, the snowpack was set as wet.

The core principle of retrieving snow depth from passive microwave remote sensing data is that snow particles scatter the microwave signals emitted from the ground, and the brightness temperature of ground declines as it crosses the snowpack. The higher the frequency, the greater the radiation scatters, and more snow particles lead to a larger brightness temperature

gradient. Therefore, the spectral gradient, namely the brightness temperature difference between lower frequency and higher

frequency is used to derive snow depth. Based on modelling and observation, the 18 GHz (K band) and 36 GHz (Ka band) are the best frequencies for deriving snow depth (Chang et al., 1987; Kelly et al., 2003). The brightness temperature difference between these two frequencies (TBD) has a good relationship with snow water equivalent.

However, frozen soil and cold desert also scatter radiation, and their existence leads to a positive TBD (Grody and Basist, 1996). Therefore, before retrieving snow depth, snow cover must be identified from other scattering sources. In this study, a modified global snow identification method is used to retrieve snow cover using AMSR-E brightness temperature. The criteria are described as following:

Cold desert: $TB_{18V}-TB_{18H}>=18$ (K) AND $TB_{18V}-TB_{36V}<=10$ (K) AND $TB_{36V}-TB_{85V}<=10$ (K)

Frozen soil: $TB_{18V}-TB_{18H}>=8$ (K) AND $TB_{18V}-TB_{36V}<=2$ (K) AND $TB_{36V}-TB_{85V}<=6$ (K)

Snow depth (cm) $= 0.7*(TB_{18H}-TB_{36H}-5)/(1-0.5f)+offset$

where, offset was monthly data which was used to decrease the influence of snow characteristics growth. They are -4.18, -3.58, -0.29, 2.15, 3.31 and 3.8 for Oct, Nov, Dec, Jan, Feb, Mar and Apr, respectively (Che et al., 2008). These criterions were developed based SSM/I (Grody and Basist, 1996). Based on the inter-sensor comparison between SSM/I and AMSR-E in Dai and Che (2010), and between AMSR-E and AMSR2 in Du et al., (2014), the brightness temperature from these sensors are close to each other. Therefore, in this study, these criterions were also applied for AMSR-E and AMSR2.

## 2.3 Meteorological station observations of snow depth

Daily snow depths and snow water equivalents were observed at 109 meteorological stations across the QTP with a spatial distribution provided in Fig. 1. Snow depths from meteorological stations were observed daily at 8:00 am using rulers, and the record is the mean value of three individual measurements. On the fifth, tenth, fifteenth, twentieth, twenty-fifth and the last day of a month (nearly every five days), snow water equivalents were measured using snow tube with a cross-sectional area of 100 cm$^2$ at the same time of snow depth measurement. And the record is also the mean value of three individual measurements.

## 2.4 Field experiments

From 20 November to 7 December, 2013, snow depths were observed along an observation route (Fig. 1, red line). During this period, little snow accumulated; only some patchy snow was distributed which cannot be measured by ruler. From 23 March to 31 March, 2014, snow characteristics were observed along an additional observation route (Fig. 1, blue line). Snow depth was recorded every 5~10 km in the snow-cover area, and snow depths in the transition region were also measured. During this field campaign, 56 snow depths were recorded. From 6 to 25 May, 2014, snow depths were observed along an additional observation route (Fig. 1, green line). During this period, there was no snow distribution except at the tops of mountains, which was not measurable.

The Binggou watershed in the Qilian Mountains, an area of 30 km$^2$, is located in the northeast of the QTP (Fig. 1, pink polygon), where dense snow depths were measured during the watershed allied telemetry experimental research (WATER) field campaign carried out in March of 2008. During this experiment, 51 snow depths were measured using snow stakes on the 2, 4, 9, 16, 19, 21, 23, and 29 of March and the 1 and 6 of April. On 29 March, 2008, airborne microwave radiometry experiment was carried out, providing brightness temperatures at the 18 and 36 GHz, and 78 snow pits including snow depth, snow density and grain size were observed at four sampling sites (Li et al., 2009; Che et al., 2012). These data were all used to evaluate the identification of snow cover by passive microwave and the accuracy of the satellite-derived snow depth.

## 3  Evaluation methods and results

The MODIS snow cover fraction product, meteorological station observations, and field campaign snow depth observations are compared with the AMSR-E/AMSR2 snow cover, and snow depths observed at meteorological stations and field experiments are compared with AMSR-E/AMSR2 snow depths.

## 3.1 Comparison with MODIS snow cover fraction product

Based on the snow-cover identification algorithm described in section 2.2, the AMSR-E brightness temperatures were used to calculate the TBD, which represents snow depth. MODIS snow cover fractions (SCF) with a resolution of 500 m were resampled to 0.1 °, similar to the AMSR-E resolution across the QTP. For every AMSR-E grid, SCF was recalculated based on the no-cloud MODIS grids (new SCF), and the number of cloud-cover grids in every AMSR-E grid was also recorded.

The AMSR-E TBDs were grouped into 5 groups: <=5 K, 5~10 K, 10~15 K, 15~20 K, >20 K, and the new SCFs were divided into six groups: <10 %, 10 %~30 %, 30 %~50 %, 50 %~70 %, 70 %~90 %, >90 %. The frequencies of each SCF with cloud fraction less than 10 % for each TBD group from 2003 to 2007 were computed, which was called TBD-SCF table. Based on the TBD-SCF table, the probability was calculated, which was the ratio of frequency of a certain group of SCF and

the frequency of all SCF with cloud fraction less than 10 %. The flowchart for building the TBD-SCF table is provided in Fig. 2.

The probabilities of all SCF groups to all TBD groups were depicted in Fig. 3. Left figures described the spatial distribution of probabilities of SCF >10 % when TBDs were more than 20 K (Fig.3 a), between 15 and 20 K (Fig. 3 b), between 10 and 15 K (Fig. 3 c), between 5 and 10 K (Fig. 3 d), and less than 5 K (Fig. 3 e). Right figures were the statistic results of different

probabilities for all group of SCF all over the QTP. The first groups with horizontal axis labelled by "> 10%" were the statistic result of the right figures. The red bar means the number of pixels with probability of "SCF >10%" between 0 and 0.1 all over the QTP, the yellow bar for between 0.1 and 0.5, the light blue bar for between 0.5 and 0.8, and the dark blue bar for more than 0.8. The other groups which labelled by ">30%", ">50%", ">70%", and ">90%" presented the same meaning corresponding their SCF range as "SCF >10%", but their spatial distribution were not presented in figures.

If SCF > 10 % was considered as snow cover, grids with TBD more than 20 K showed 4.9 % snow-free area, 82.9 % snow area, and 12.2 % uncertainty area. The uncertainty areas included 6.1 % high possibility of snow cover area and 6.1 % high possibility of snow-free area. A decrease in TBD causes the certainty ratio to decline and the uncertainty to increase. TBD between 15 and 20 K showed 5.9 % snow-free area, 68.2 % snow-covered area, and 25.9 % uncertainty area. The TBD between 5 and 10 K presented the highest uncertainty. When the TBD is less than 5 K, the QTP is dominated by no snow,

and the snow-covered areas are mainly glaciers and lake ice, based on the land cover map. However, there is a large area of uncertainty with a low possibility of snow. Therefore, snow cover is difficult to be identified when the TBD is between 5 and 15 K. Although there was large overestimation over the QTP, if one threshold of TBD was used to identify snow cover over the QTP, "TBD >5 K" was best.

If SCF >30 % is considered as snow cover, the uncertainty areas increase when TBD is more than 5 K, but the snow-free

areas increase when TBD is less than 5 K. If SCF > 50 % is considered as snow cover, only 3.3 % of the area is definitely

identified as snow when TBD is between 5 and 10 K, 9.8 % when TBD is between 10 and 15 K. With an increase in TBD, the snow cover areas increase, and the uncertainty area increases. Therefore, although there is no obvious relationship between TBD and snow cover fraction, TBD can reflect snow cover fraction to a certain extent.

With SCF > 0.1 as snow cover and TBD > 5 K is the threshold to identify snow from AMSR-E, the overall accuracy, underestimation, overestimation, commission and omission errors of AMSR-E were analyzed (Fig. 4, Table 1). The overall accuracy is 66.7 % and varies spatially. In this condition, 27.6 % of snow-free areas are misclassified as snow cover (commission), and 47.4 % of snow cover grids are not be detected by AMSR-E (omission); meanwhile, 56.1 % of grids identified as snow covered by AMSR-E were free of snow (overestimation), and 21.1 % of snow-free grids from AMSR-E were in fact covered by snow (underestimation), which is mainly distributed in the lake and glacier areas. The lowest accuracy occurs in the northwest area of the QTP, where commission error reaches up to 0.6-0.8. Although the overall accuracy for the cold desert areas is more than 0.8, in most of these areas, the omission error is also up to 0.8, which means that 80 % of snowfall cannot be detected by AMSR-E. In these areas, snowfall is a rare event, and snow depth is shallow, which changed TBD slightly. If pixels with SCF >0.3 is assumed to be snow pixel, the overestimation, underestimation, commission and omission errors are 72.2%, 0.9%, 17.5% and 9.9%, respectively. If SCF > 0.5 is the threshold between snow and no snow, they are 83.2%, 0.4 %, 19.5% and 7.2%, respectively. The overestimation and commission errors increase with the increase of the threshold, and the underestimation and omission errors decrease. Therefore, the high overall accuracy of these areas is due to the large number of snow-free days. In the mountainous areas of southeast and northeast Qilian and the northwest area of the QTP, AMSR-E showed high overestimation and commission errors.

Generally, based on MODIS SCF, snow cover accuracy of AMSR-E across the QTP varies spatially, and showed the overall accuracy of over 60 % in most of areas. But it was also characterized by large areas of overestimation and omission errors.

### 3.2 Comparison with observed snow depth

Daily snow depths from meteorological stations from 2003 to 2007, snow depth from the three observation routes in 2013 and 2014, and snow depth observed at the Binggou watershed in 2008 were compared with the snow cover and snow depth derived from AMSR-E or AMSR2.

### 3.2.1 Comparison with meteorological station observation

Snow depths were derived from AMSR-E at grids that meteorological stations located on, and then were compared with station records. The comparison results showed that the overall accuracy of AMSR-E snow cover is 82.7 %, where 41.6 % of snow covered grids were not detected by AMSR-E, and 16.0% of snow-free grids were misclassified as snow covered by

AMSR-E. The overestimate and underestimate are 84.3 % and 2.5 %, respectively. A meteorological station may not represent the status of an entire PMW grid in the complex territorial region, therefore snow cover fractions in the PMW grid were derived based on MODIS snow cover production and compared with meteorological observations. The results showed that when MODIS SCF was greater than 10 %, only 22.4 % of snow depth observations were greater than 0 cm, a MODIS SCF greater than 30 % corresponded to 39.8 % of observations greater than 0 cm, and a MODIS SCF greater than 50 %

corresponded to 54.9 % of observations greater than 0 cm. Therefore, although station snow observations are in good general agreement with the snow cover MODIS grid (Yang et al., 2015), they cannot represent the snow cover in a PMW grid across the QTP.

Due to the disagreement between the PMW grid and station-based snow cover measurements, snow depths from stations and AMSR-E greater than 0 were compared (Fig. 5). The results showed that the correlation coefficient between them was 0.124,

and  AMSR-E overestimates snow depths across the QTP, in agreement with results of Yang et al. (2015). The mean snow depth, bias and RMSE are 4.0 cm, -0.45 cm and 6.7 cm, respectively, and the relative error is 131.4 %. From Fig. 5, snow depths greater than 20 cm were always underestimated by AMSR-E, caused primarily by the data that came from the Nyalam station (Id: 55655, Latitude: 28.18 N; Longitude: 85.97 E) located in the Himalaya Mountains. If the data at this station are removed from the statistics, the mean snow depth, the bias and RMSE are 3.5 cm, 1.7 cm and 5.5 cm, respectively,

and the relative error is 152.3 %.

Therefore, snow depth derived from AMSR-E showed low consistency with that from stations over QTP. Because of the complex terrain, meteorological stations, mainly distributed in the valley with small snow and no continuing accumulation period, and have no obvious tendency of increase or decrease, may not present the snow status of a PMW pixel.  The "Former Soviet Union Hydrological Surveys" (FSUHS) presented highest station density (approximately one transect per

100 km gridcell) and is primarily composed of non-complex terrain with maximum elevation differences of <500m. When

compared with these station observations, PMW presented high correlation with in situ data, although underestimated the snow depth in the Former Soviet Union (Armstrong and Brodzik, 2002). Therefore, it is the complex terrain and special distribution of snow cover that make the meteorological station observation lack of their meaning on evaluation of snow depth across the QTP.

**3.2.2 Comparison with field observations**

Observations from December of 2013 and May of 2014 indicated sparse snow along the observation route, a result also shown by AMSR2. During the observations in March of 2014, 56 points of snow depth were measured within 33 AMSR2 grids (Figure 1). Comparison between ground observations and retrievals from AMSR2 indicates that the retrieval accuracy of snow cover from AMSR2 is 94 %. The average snow depth of observed measurements is 6.71 cm, the bias between them is 0.27 cm, RMSE is 5.4 cm, and the correlation coefficient is 0.574 (Fig. 6 a). According to MODIS fractional snow cover products, snow cover fraction of the pixels that these points located in ranged from 0.5 to 0.9 when the observations showed snow. If the snow cover fraction is considered in the comparison, the bias is 1.77cm, RMSE is 5.66 cm, showing general overestimation. Therefore, in these observation areas, snow cover can be detected accurately by PMW which was consistent with the comparison between MODIS and AMSR-E, but snow depth was overestimated.

In 2008, there were five groups of snow depth observations and a total 51 points, all within an AMSR-E grid in the Binggou watershed (Che et al., 2012). The average snow depths of the 51 points for the 2, 4, 9, 16, 19, 21, 23, and 29 March and 1 and 6 April were 18.2 cm, 15.5 cm, 21.5 cm, 20.0 cm, 24.6 cm, 21.5 cm, 24.2 cm, 18.0 cm and 14.4 cm. Snow depths varied between 0 and 60 cm. Compared with these samples, the snow depths derived from AMSR-E generally present underestimation; the bias is -10.0 cm, and the RMSE is 10.5 cm (Fig. 6 b). Therefore, based on the field investigation, snow cover can be detected accurately by AMSR-E because of thick snowpack, but the accuracy of snow depth retrieval is low. Although the observations in the Binggou watershed were dense, due to the large spatial variation in snow depth and topography, an average snow depth may not represent the snow depth of a whole grid. Che et al. (2008) analyzed the relationship between snow depth distribution, elevation, and directional aspect using the snow depth estimated from airborne radiometer data with a footprint of 16-39 m at 36 GHz and 158-395m at 18 GHz. The authors found that snow cover was primarily distributed in a northerly aspect. The snow cover fractions across the QTP derived from the MODIS snow cover

product are 52 %, 35 %, 45 %, 34 %, 36 %, 46 %, 42 %, 17 % and 21 % for 2 March, 4 March, 9 March, 16 March, 21 March, 23 March, 29 March, 1 April and 6 April, respectively, and the average snow depths in the AMSR-E footprint are calculated by multiplying the snow cover fraction by the observed mean snow depth. The average snow depths in the AMSR-E footprint are compared with the derived snow depth, exhibiting average snow depth, bias, RMSE, and absolute

relative error of 7.4 cm, -0.4 cm, 2.2 cm, and 29.5 %, respectively (Fig. 6 b).

Therefore, the spatial inhomogeneity of snow depth causes the difference between satellite and in situ observation. Evaluation of PMW snow depth over the QTP requires dense sampling in a whole pixel.

As a whole, accuracy of PMW snow cover in the QTP presented a spatial heterogeneity, and the difference of snow depth between estimation and in situ observations also varied with observation data.

**4. Sources of error**

Although both existing research and this study reported the overestimation phenomenon over the QTP, the causes were still in suspense. Besides, this study found there was also serious omission problem in the shallow snow areas, and difference of snow depth between the estimation and in situ data. Therefore, we discuss potential reasons for the misclassification and bias in this section.

**4.1 Cold desert**

Based on the classification criterion of Grody and Basist (1996), cold desert presented large polarization. There are large areas of cold desert on the middle and Northwest part of QTP, which also showed scattering features. The omissions mainly appeared in these areas, with the exception of the lake ice areas. In these areas, there is no heavy snow, and the snow depth is usually less than 5 cm. The fallen snow melts quickly in a few days, resulting in a small TBD change. Take the Tuotuohe

station (Id:56004, Latitude: 34.22N, Longitude: 92.43E, Fig. 1) which located on these areas (Fig. 1), for example; during the winter, ground scatters the microwave signal and presents weak scattering features. The TBD contributed by ground is less than 5 K, but even if the cold desert was covered by snow, the TBD did not increase and remained less than 5 K. Liquid water melted from snow cover will even decrease the TBD. The criterion for cold desert identification presented in the section 2.2 removes not only the desert as a scatter but also the snowpack. If the criterion is not used, AMSR-E will seriously

overestimate the snow cover. The Gobi desert areas in the middle and south of the QTP showed the same phenomenon, but the northwest of the QTP showed the opposite phenomenon, where there is little snow, but PMW classify them as large area of snow.

## 4.2 Soil temperature

$TB_{36V}$ is sensitive to topsoil temperature (Holmes et al., 2009; Zeng et al., 2015). Statistical analysis between TBD (K) and $TB_{36V}$ at 109 stations showed that TBD has a significant negative correlation with $TB_{36V}$ (Fig. 8 a) at the confidence level of 0.95, but no obvious relationship with snow cover fractions. Batang station (Id: 56247, latitude: 30.00N, Longitude: 99.10E, Fig. 1) is a typical station, where snowfall is rare, the PMW grid of this station was seldom covered by snow, and the snow cover fraction in the AMSR-E grid was greater than 10 % on only a few days based on MODIS snow cover fraction products.

The temporal variation in $TB_{36V}$, TBD, and snow depth at this station also indicates that a decrease of $TB_{36V}$ is accompanied by a TBD increase to over 5 K with a snow depth of 0 cm (Fig. 8 b). $TB_{36V}$ and TBD have a highly negative correlation (Fig. 8 c). Therefore, the ground temperature is also a main reason causing large TBD. Based on the MODIS land surface temperature, the land surface temperature in the Northwest of the QTP showed lowest value, where the overestimation was most serious (Fig. 4).

The penetrability of 18 GHz and 36 GHz are different and depend on the soil features. In the summer, the brightness temperature at 18 GHz and 36 GHz is emitted from the ground surface, but with decrease of temperature and soil freezing, the penetration depth of 18 GHz is larger than the 36 GHz. The higher temperature at deeper place contributes to the brightness temperature of the 18 GHz and lower temperature close to the surface contributes to the brightness temperature of the 36 GHz. Besides, the 36 GHz is sensitive to both ground surface temperature and snowpack, but ground surface temperature is also influenced by snowpack. Because of snowpack thermal insulation and thermal transfer of soil, ground

surface temperature may stay high when covered by snow. As the brightness temperature of the 36 GHz emitted from ground increases, it is also reduced by snowpack when arriving at sensor. Therefore, it is difficult to discriminate what is the main factor to cause the decrease of brightness temperature at 36 GHz.

Therefore, we believe the ground feature is the main resource of errors. Accurately modelling the brightness temperature of different bands emitted from the ground is the key to improve the accuracy of snow cover detection (Jiang et al., 2007 and 2011).

## 4.3 Atmospheric correction

Thinner atmosphere across the QTP was the hypothesized cause of overestimation of snow depth from PMW remote sensing (Savoie et al., 2009; Qiu et al., 2009). Prior researchers assumed that general algorithms built based on satellite brightness temperature and ground snow depth implicitly accounted for the presence of an atmosphere. In this study, we used the atmosphere correction method developed in Savoie et al. (2009) to adjust the brightness temperature of QTP to that of a lower elevation and then derive the snow cover from AMSR-E from 2003 to 2007. The derived snow cover was compared

with snow cover fraction estimates from MODIS. The comparison results indicated that the overall accuracy improved from 66.7 % to 72.2 %, the commission error decreased from 27.6 % to 14.2 %, and overestimation error decreased from 56.1 % to 46.8 %, but the omission error increased from 47.4 % to 60.8 %, meaning that an additional 13.4 % of snow cover was not detected (Table 2). If the TBD threshold used for identifying snow cover changed to 1 or 2 K, then the overall accuracy, overestimation and omission would exhibit the same change in trend as with an atmospheric correction.

## 4.4 Spatial resolution and topography

The footprint of airborne radiometer data in the Binggou watershed experiment were 16-39 m at 36 GHz and 158-395 m at 18 GHz. Considering the speed of the aircraft and interval time of radiometers, the brightness temperatures of both frequencies were gridded at 90 m resolution. The observed points were distributed in separate grids. Che et al. (2008) used an MEMLS model to simulate the brightness temperature of snow cover for each observation point and developed a snow

depth retrieval algorithm in the Binggou watershed. The mean absolute and relative errors of snow depth estimates were approximately 3.5 cm and 14.8 % for the stake and sampling-site regions. The mean absolute and relative errors for AMSR-E are 2.0 cm and 29.5 %, respectively, in the AMSR-E grid. Although the derived snow depths from airborne and satellite radiometry agreed with each other, the average airborne brightness temperature and AMSR-E brightness temperature at 36GHz presented a large bias.

The satellite and airborne radiometers have similar radiation characteristics and were all well calibrated. The aircraft flew at an altitude of 5000 m, where atmospheric influence on the airborne and satellite brightness temperatures should be the same. The difference between the airborne and satellite data is the spatial resolution, overpass time and incidence angle. In the Binggou watershed, snow cover presented strong heterogeneity. Fifty-one snow stakes covered fifty-one airborne grids located on seven MODIS grids which only overlapped with a small part of the PMW grid (Fig. 9). Fifty-one snow depths varied between 0 and 60 cm, which can be detected by airborne radiometry, but for MODIS, they were all covered by snowpack. For the AMSR-E grid, they did not reflect snow distribution, although they were measured in different directional aspects and elevations.

Airborne experiments were carried out in daytime, which was closer to the ascending overpass of AMSR-E. The ascending TBD was less than 2 K, and the descending TBD was approximately 11 K, as presented in Fig. 9. In daytime, the snow cover melted in some areas, which led to spatially variant liquid water content and likely caused some of the differences between the airborne and satellite brightness temperature. In addition, the scan areas of the airborne radiometry were not identical to the satellite observations, which is an additional cause of the large gap between the airborne and satellite brightness temperatures for heterogeneous distribution of snow cover in the Binggou watershed.

**4.5 Snow characteristics**

Based on spectral gradient algorithms, derived snow depths are closely related to TBD. However, TBD is not only influenced by snow depth but also other snow characteristics, in particular, snow grain size. At the beginning of snowfall, snow grain size is small and the snowpack is transparent for microwave, so passive microwave remote sensing underestimates the snow depth in this period. With increasing snow age, grain size grows, which contributes to TBD, so snow depth may be overestimated by passive microwave remote sensing. Although the soil temperature and the land type were the main causes of errors in the QTP, the instant snow could not be detected for the extremely low scattering of small grain size. Therefore, accurately monitoring the snow depth using passive microwave requires a priori knowledge of snow characteristics (Dai et al., 2012; Che et al., 2016; Huang et al., 2012; Tedesco and Narvekar, 2010). In this study, 16 % of snow depths greater than 10 cm observed at meteorological stations were misclassified as snow-free grids by AMSR-E. This misclassification occurred in the areas of sparse snow, where heavy snowfall occurred occasionally but melted in 1-3 days.

During the field campaign in March 2014, snowpack measured on 23 March was fresh snow but was misclassified as no snow cover.

Therefore, accurately modelling the ground brightness temperature at both frequencies and snow characteristics are two key factors for improving snow depth and snow cover accuracy of PMW. However, the strong heterogeneity of snow distribution

over the QTP requires a retrieval algorithm with high resolution.

**5 Discussions**

Although satellite-based passive microwave brightness temperature data have been used to monitor global and regional snow depth since the 1980s, there were still some uncertainties on the snow depth retrieval algorithm in the QTP. Based on existing research on the evaluation of PMW products, forest and grain size were the main causes resulting in the low

accuracy of PMW algorithm. In the forest regions, snow depth was usually underestimated by PMW, and many methods had been developed to overcome it (Forest et al., 1997; Vander et al., 2015; Pullianen et al., 1999; Che et al., 2016). In the QTP, forest mainly distribute in the southeast region with rare snow, therefore, it is not the dominant factor in the QTP. However, in the Fig 4 b, the northeast region with omission errors was the main forest areas. In this area, snowfall was a rare event, and volume scattering signal was weak, plus the forest cover, it was difficult for PMW to detect snow.

Large grain size of snow can scatter much more irradiance than small one; therefore, fresh snow cover with small grain size tends to be underestimated, while snow cover with large grain size (e.g. depth hoar) tends to be overestimated due to its strong scattering. In order to solve this problem, the grain size growth model was developed by Kelly et al. (2003), and the a priori snow characteristics was used in the snow depth retrieval in northwest and northeast of China (Dai et al., 2012; Che et al, 2016). The Globsnow snow product used the assimilation method to optimize grain size, voiding the measurement of

grain size (Pullianen, 2006). For the QTP, most areas were characterized by shallow snow or instant snow. Fresh snow melted out in few days resulting in weak scattering which is difficult to be detected by PMW, therefore, snow cover was underestimated in these areas. However, due to the complex topography, snow accumulated in cold and shady areas can survive for a long time and its grain size increased with the metamorphism and form the depth hoar, resulting in strong scattering and then causing the overestimation. Therefore, because of the lack of efficient grain size data in the QTP, the

accuracy of estimated snow depth was certainly influenced.

However, in this study, we found that except for the problem of forest and grain size, there were other some special problems for QTP which were the large areas of cold desert and frozen soil, and the patchy snow cover. In the section 4, we analysed the overestimation in the cold desert and frozen soil areas. There were some studies presenting the scattering features of cold desert and frozen soil, they were all of weak scatters, and the TBD at vertical polarization caused by desert and frozen soil were less than 10 K and 2 K, respectively (Grody and Basist, 1996). The criterion has been used to remove the other scatters in the global algorithm, and it works in most regions. But in the QTP there are still large areas presenting overestimation after using this criterion (Fig. 4).

Some research reported that the overestimation came from the atmosphere (Savoie et al., 2009), but based on the analysis in this study, the atmosphere correction decreased the commission errors and improved the general accuracy, but sacrificed the omission errors. This study also showed that the TBD was mainly controlled by soil temperature. It has a strong negative correlation with brightness temperature at 36 GHz for vertical polarization which is the most sensitive to ground surface temperature. Brightness temperature at 36 GHz is much more sensitive to the land surface temperature than 18 GHz. With the surface temperature declines, the brightness temperature at 36 GHz decreases quickly. But the brightness temperature at 18 GHz keeps stable, because it is influence by temperature at deeper layer of soil. Therefore, brightness temperature at 36GHz is lower than that of 18GHz, and the difference between them increases with the decrease of temperature of surface soil. Moreover, because of the freeze/thaw cycle of surface soil, the frozen soil becomes incompact and dry. The fine-scale soil and sand particles are scatters which also weaken the brightness temperature at 36GHz (England, 1976). In the northwest of the QTP, the surface temperature is very low and the polarization difference larger than 30 K which is the characteristics of desert. Therefore, we inferred that the combined action of frozen soil and desert resulted in large TBD, and then caused the serious overestimation.

Furthermore, patchy distribution of snow cover in the QTP was another cause for uncertainty of PMW with coarse resolution. For the high latitude regions, snow is of large-area phenomenon, PMW works well to detect snow cover. But for the QTP which is characterized by low latitude and high altitude, not only in the mountainous areas snow cover distributes in inhomogeneity, but also on the plain areas. In a PMW pixel, both snow particles and low-temperature bare soil produces TBD, which certainly results in overestimation. The patchy distribution does not only bring problem to the derivation of

snow depth from PMW, but also to the evaluation of the snow depth in a PMW pixel. In this study, we used the MODIS snow cover to evaluate the accuracy of snow cover, and used station point data, sampling in lines and intense sampling data to assess the accuracy of snow depth. And we found that it is not so reasonable to use station observations to evaluate the accuracy of PMW snow depth because of weak representation of many stations, and the general accuracy cannot depict the

accuracy of PMW snow depth in the QTP, neither can the simple overestimation. Therefore, it is necessary to develop a retrieval algorithm for improving the spatial resolution of snow depth.

## 6  Conclusions

This study presented the accuracy of snow depth product derived from PMW by comparing with MODIS snow cover fraction and in situ data, and analysed the potential causes resulting the uncertainties of the product.

The results showed that the overall accuracy of snow cover derived from PMW remote sensing across the QTP varies spatially, based on MODIS snow cover fraction. Commission errors were mainly distributed in the northwest and southeast where ground temperature was low, and omission errors were found in the cold desert areas with sparse snowfall. The overestimation and commission errors decreased with the MODIS SCF, and underestimation and omission errors increased. The AMSR-E/AMSR2 snow depth was compared with the observations at meteorological stations and field investigation

presented that snow depth in the field investigation showed higher consistency with estimated ones than meteorological station observations. Most of stations locate on the low land, and cannot represent the snow depth in a PMW pixel. When compared with MODIS snow cover, snow cover at stations always less than the MODIS observations. Therefore, not all station observation can be used to evaluate the accuracy of PMW snow depth.

Low ground temperature is the dominate reason causing overestimation of snow cover by PMW. Instant snow cover with

small grain size led to the omission errors in the shallow snow areas. The mountainous topography and the coarse resolution of PMW resulted in the large disagreement between the snow depth derived from AMSR-E and in situ observations or airborne radiometry. Therefore, accurately monitoring the spatiotemporal distribution of snow depth across the QTP requires improving the retrieval accuracy of PMW as well as the spatial resolution. A new snow depth retrieval algorithm is suggested to combine optical remote sensing of snow cover, land surface temperature product and PMW.

## Acknowledgements

This study was supported by the National Natural Science Foundation of China (91547210, 41401414 and 41271356), the China State Key Basic Research Project (2013CBA01802), and the Chinese Academy of Sciences Project (KJZD-EW-G03).

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

# List of Table captions

**Table 1 Errors in derived snow cover from AMSR-E based on MODIS snow cover fraction and meteorological stations.**

5 **Table 2 Errors in snow cover derived from AMSR-E data and atmosphere corrected AMSR-E data over the QTP, based on MODIS snow cover fraction.**

**Table 1 Errors in derived snow cover from AMSR-E based on MODIS snow cover fraction and meteorological stations (a), and corresponding confusion matrix (b).**

(a)

| | Overall accuracy | Commission | Omission | Overestimation | Underestimation |
|---|---|---|---|---|---|
| MODIS snow cover fraction | 66.7 % | 27.6 % | 47.4 % | 56.1 % | 21.1 % |
| Meteorological stations | 82.7 % | 16.0 % | 41.6 % | 84.3 % | 2.5 % |

(b)

| Confusion matrix | Snow(MODIS) | No snow(MODIS) | Snow(station) | No snow(station) |
|---|---|---|---|---|
| Snow(PMW) | 1367354 | 1749417 | 5139 | 27543 |
| No snow(PMW) | 1232973 | 4597783 | 3656 | 144368 |

Given a:snow for (MODIS or station) and PMW, b: snow for (MODIS or station) but no snow for PMW, c: no snow for(MODIS or station) but snow for PMW, d: no snow for both (MODIS or station) and PMW.

Overall accuracy=(a+d)/(a+b+c+d);  Overestimation error=c/(a+c); Underestimation error=b/(b+d)

Commission error= d/(c+d); Omission error=b/(a+b)

**Table 2 Errors in snow cover derived from AMSR-E data and atmosphere corrected AMSR-E data over the QTP, based on MODIS snow cover fraction (a), and corresponding confusion matrix (b)**

(a)

| | Overall accuracy | Commission | Omission | Overestimation | Underestimation |
|---|---|---|---|---|---|
| Original | 66.7 % | 27.6 % | 47.4 % | 56.1 % | 21.1 % |
| After atmosphere correction | 72.2 % | 14.2 % | 60.8 % | 46.8 % | 22.6 % |

(b)

| Confusion matrix | Original | | After correction | |
|---|---|---|---|---|
| | Snow(MODIS) | No snow(MODIS) | Snow(MODIS) | No snow(MODIS) |
| Snow(PMW) | 1367354 | 1749417 | 1023344 | 901632 |
| No snow(PMW) | 1232973 | 4597783 | 1586860 | 5441964 |

# List of Figure captions

**Fig. 1. Distribution of Meteorological stations, the location of Binggou watershed, and three snow observation routes described in the text overlaid on a digital elevation model for elevation and topography of the QTP.**

**Fig. 2. Flowchart for building the TBD-SCF table, which provides the relationship between snow cover fraction (SCF) and passive microwave brightness temperature difference (TBD). The SCF of the PMW grid was calculated based on the MOD10A and MYD10A products from 2003 to 2007, and TBD was computed using AMSR-E brightness temperature at 18 and 36 GHz for horizontal polarization from 2003 to 2007.**

**Fig. 3. Spatial distribution of frequency of SCF greater than 10 % across the QTP and histograms of frequency for each SCF group. (a) TBD > 20 K, (b) 15 K < TBD <= 20 K, (c) 10 K < TBD <= 15 K, (d) 5 K < TBD < 10 K, (e) TBD<=5 K**

**Fig. 4. Spatial distributions of the general accuracy (a) omission errors (b) commission errors (c) underestimation errors (d) and overestimation errors (e) of AMSR-E across the QTP.**

**Fig. 5. Scatter plot of snow depths observed at meteorological stations and those derived from the AMSR-E from 2003 to 2007.**

**Fig.6. Comparison between measured snow depth and estimated snow depth from AMSR-E/AMSR2, (a) for March, 2014 along the observation route, (b) for March, 2008 in the Binggou watershed: bar graph of observed snow depth, area-weighted observed snow depth, and estimated snow depth from AMSR-E, and line graph of snow cover fractions on different days.**

**Fig. 8. Relationship between TBD and $TB_{36V}$ at all stations (a) and Batang station (Id: 56247) (c), and the temporal variation of $TB_{36V}$, TBD, and snow depth observed at Batang station (b).**

**Fig. 9. Distribution of snow cover fraction derived from MODIS products in the Binggou watershed, locations of snow stakes set during the Binggou watershed experiment, and the brightness temperature difference between the 18 GHz and 36 GHz from AMSR-E.**

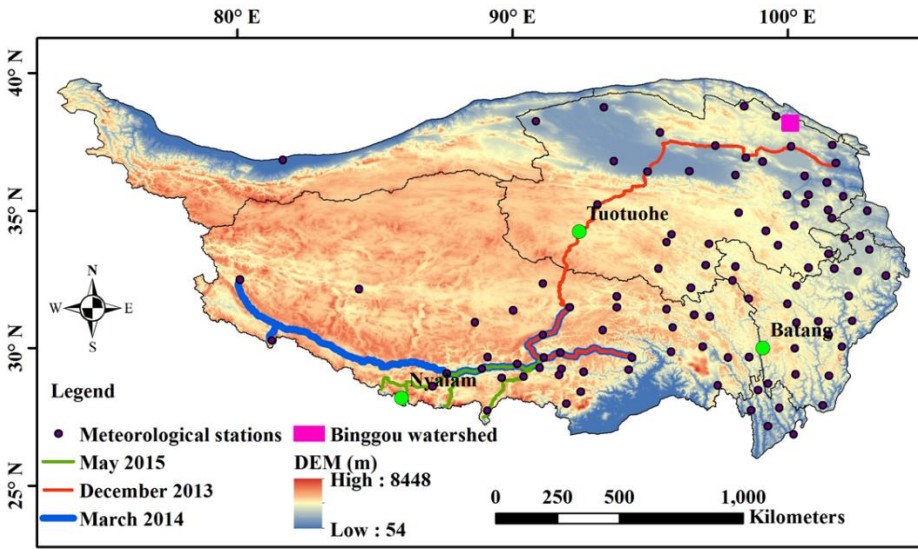

**Fig. 1. Distribution of Meteorological stations, the location of Binggou watershed, and three snow observation routes described in the text overlaid on a digital elevation model for elevation and topography of the QTP.**

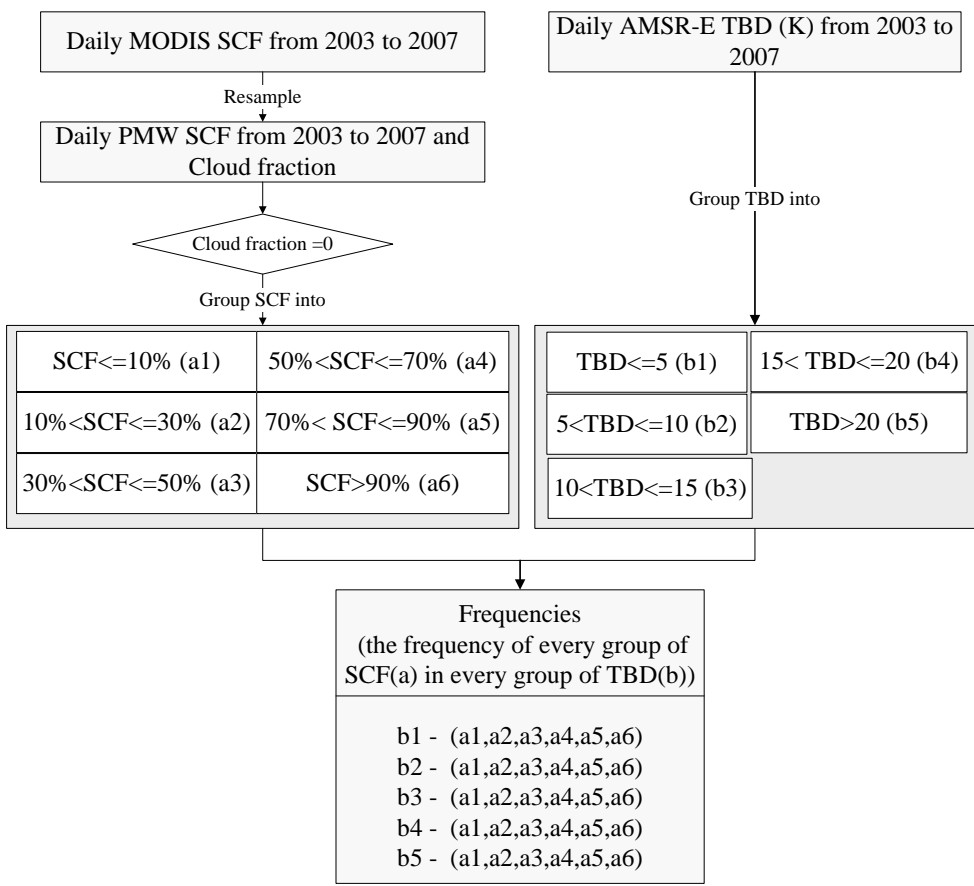

Fig. 2. Flowchart for building the TBD-SCF table, which provides the relationship between snow cover fraction (SCF) and passive microwave brightness temperature difference (TBD). The SCF of the PMW grid was calculated based on the MOD10A and MYD10A products from 2003 to 2007, and TBD was computed using AMSR-E brightness temperature at 18 and 36 GHz for horizontal polarization from 2003 to 2007.

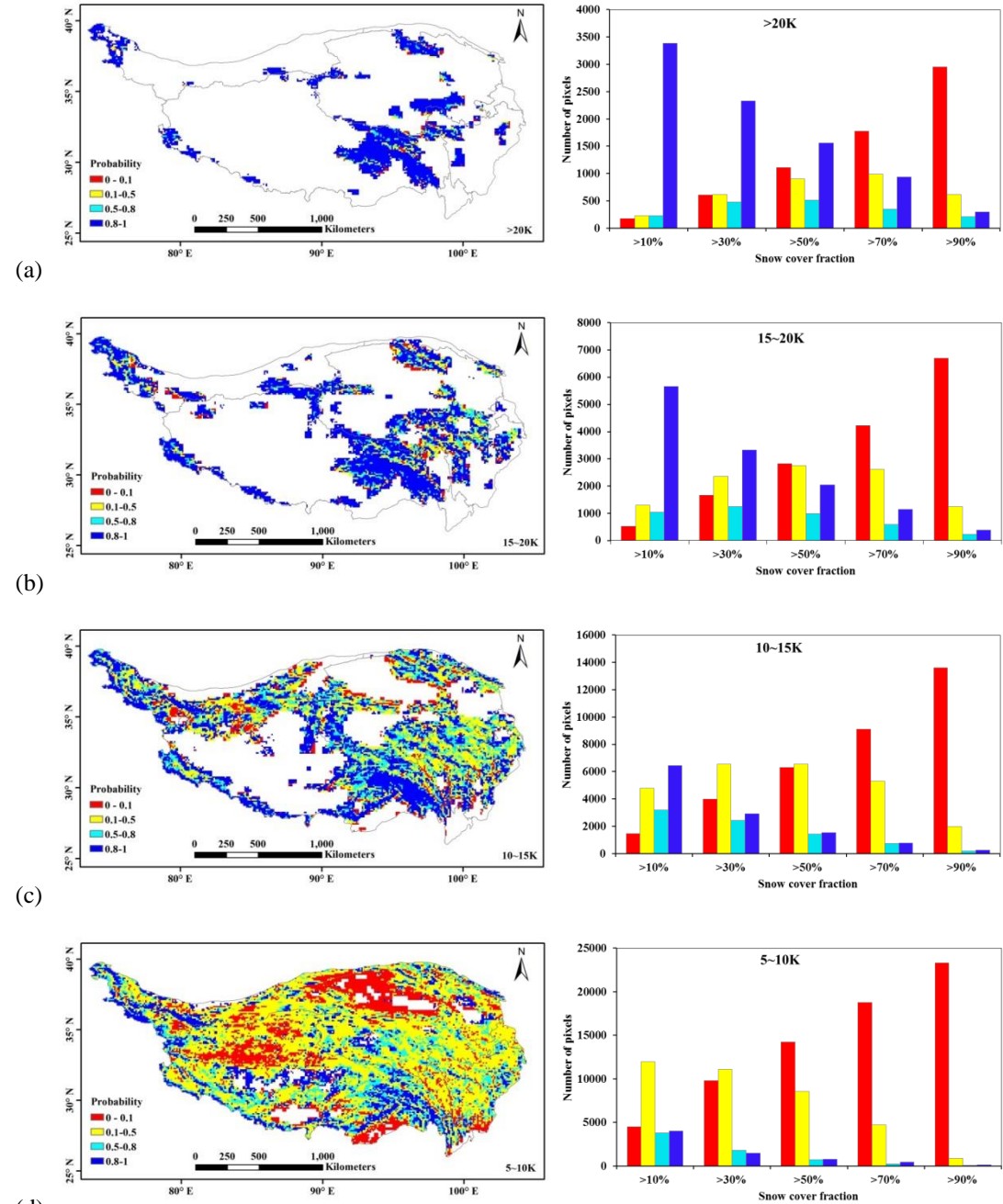

(a)

(b)

(c)

5    (d)

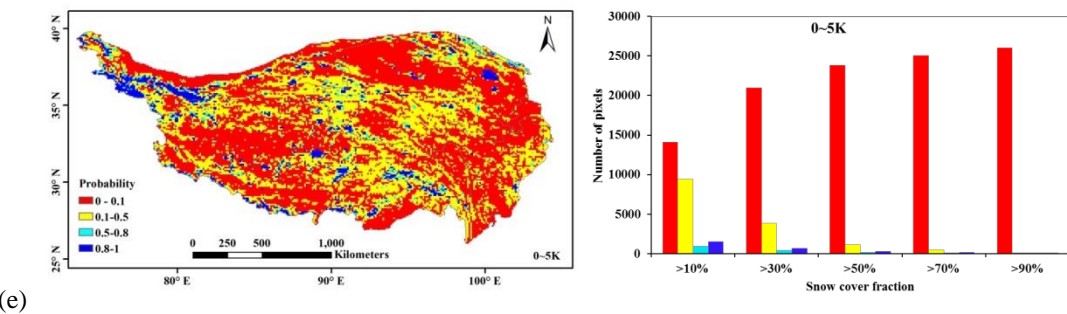

(e)

**Fig. 3. Spatial distribution of frequency of SCF greater than 10 % across the QTP and histograms of frequency for each SCF group. (a) TBD > 20 K, (b) 15 K < TBD <= 20 K, (c) 10 K < TBD <= 15 K, (d) 5 K < TBD < 10 K, (e) TBD<=5 K**

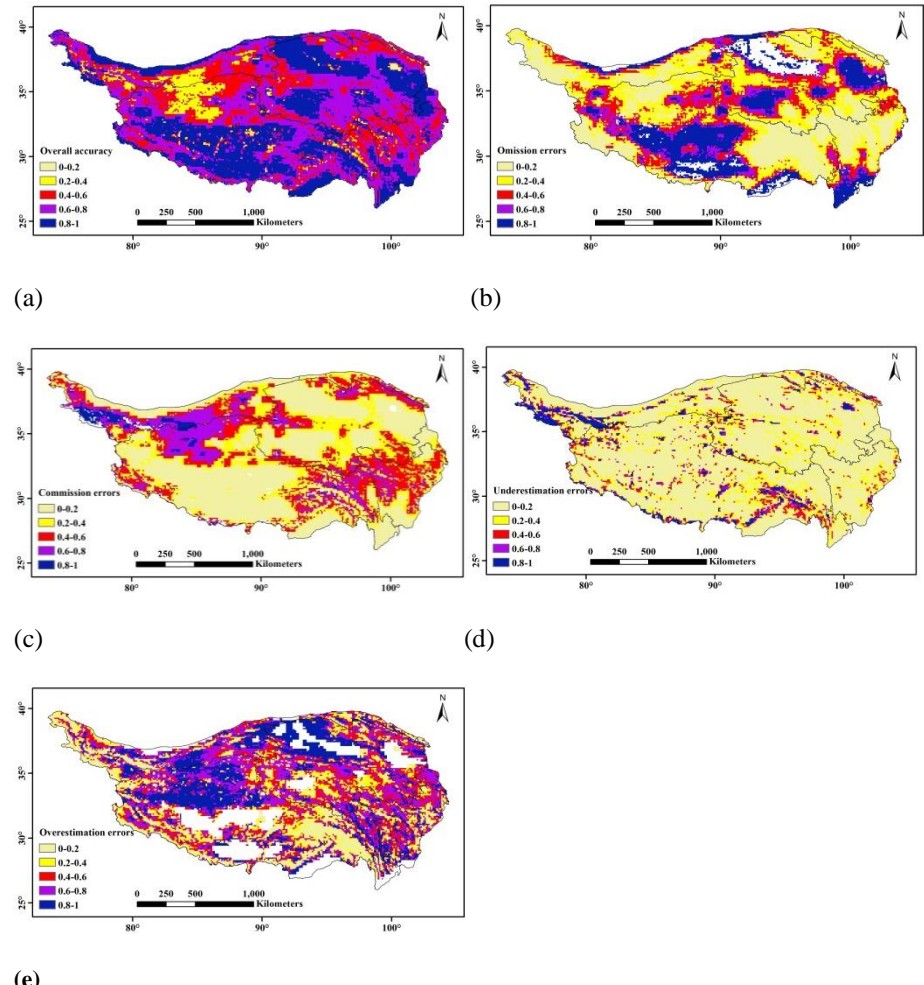

(a)

(b)

(c)

(d)

(e)

Fig. 4. Spatial distributions of the general accuracy (a) omission errors (b) commission errors (c) underestimation errors (d) and overestimation errors (e) of AMSR-E across the QTP.

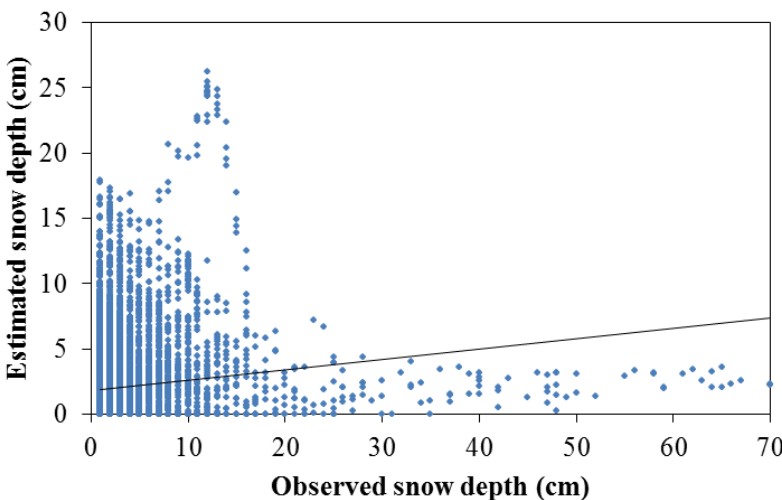

**Fig. 5. Scatter plot of snow depths observed at meteorological stations and those derived from the AMSR-E from 2003 to 2007.**

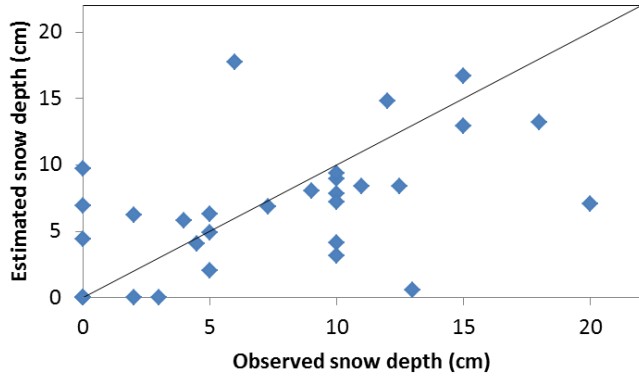

(a)

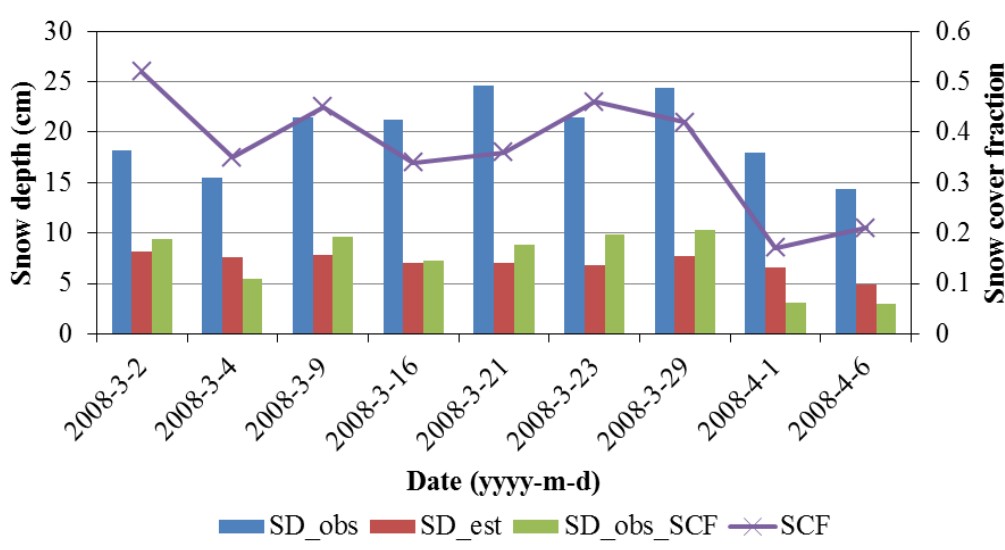

5        (b)

**Fig.6. Comparison between measured snow depth and estimated snow depth from AMSR-E/AMSR2, (a) for March, 2014 along the observation route, (b) for March, 2008 in the Binggou watershed: bar graph of observed snow depth, area-weighted observed snow depth, and estimated snow depth from AMSR-E, and line graph of snow cover fractions on different days.**

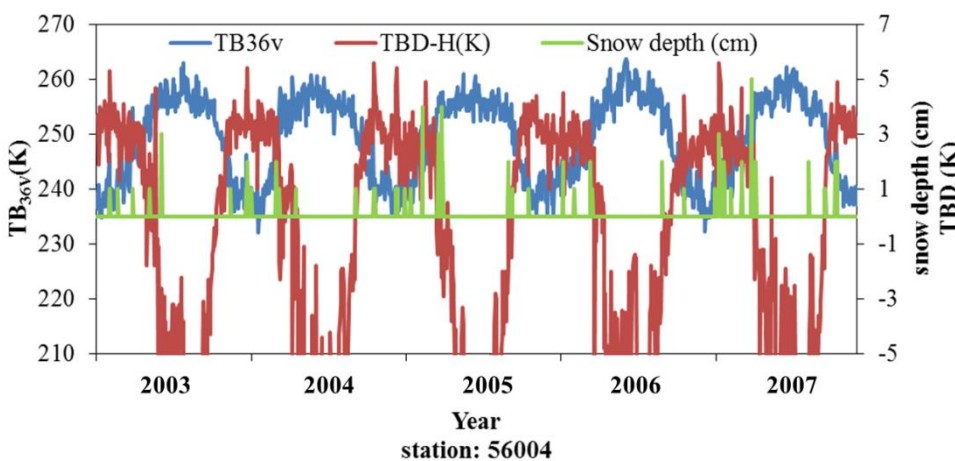

**Fig. 7.** Temporal variation of brightness temperature at 36 GHz for vertical polarization (TB$_{36V}$), TBD, and snow depth observed at Tuotuohe station (Id:56004).

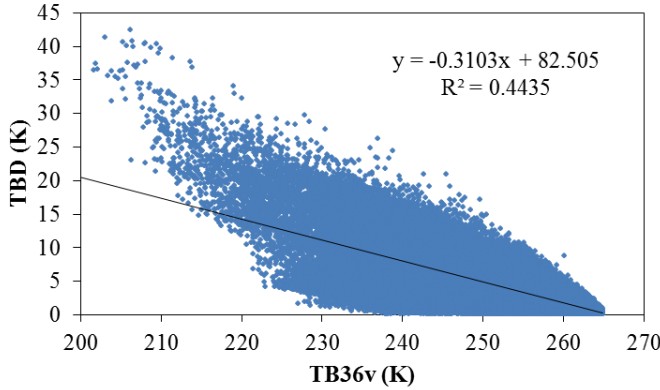

(a)

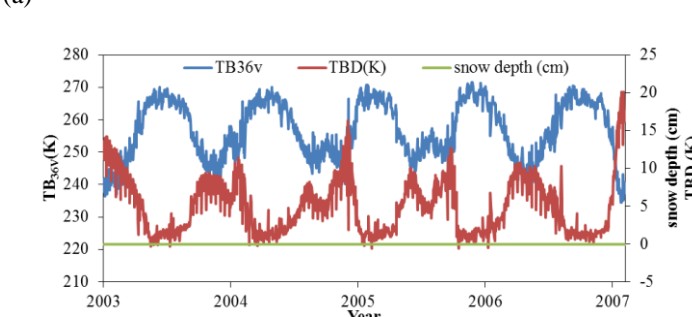

(b)

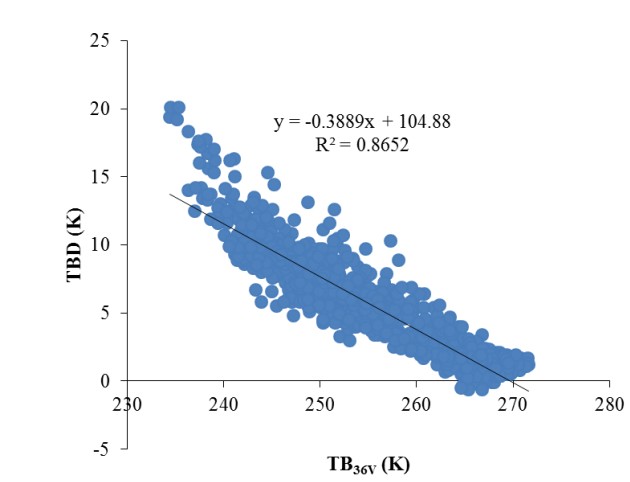

(c)

**Fig. 8. Relationship between TBD and TB$_{36v}$ at all stations (a) and Batang station (Id: 56247) (c), and the temporal variation of TB$_{36V}$, TBD, and snow depth observed at Batang station (b).**

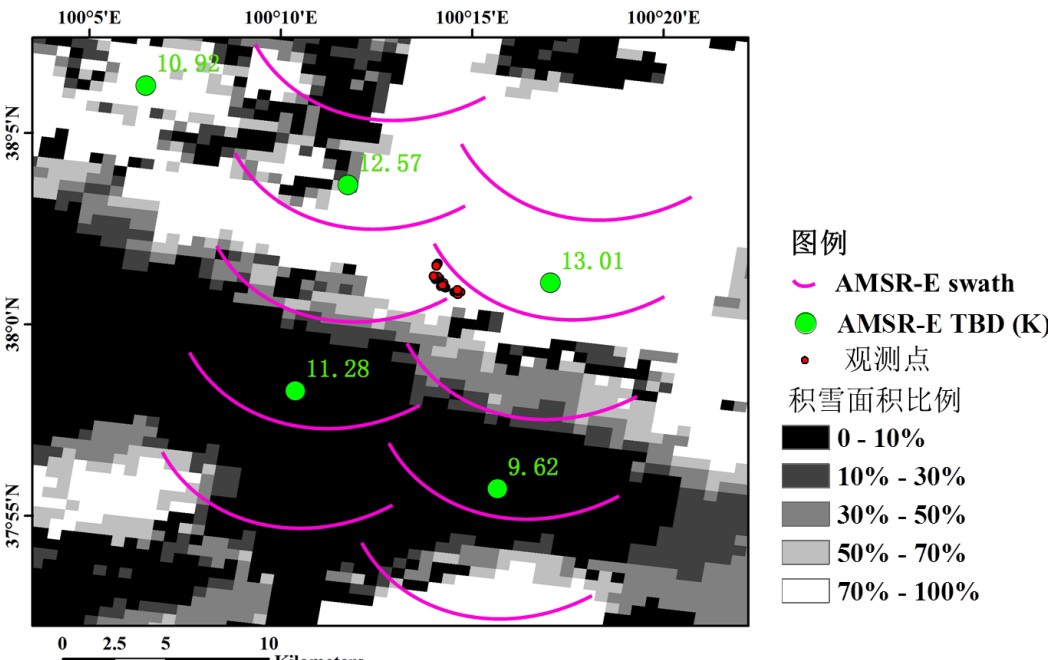

**Fig. 9. Distribution of snow cover fraction derived from MODIS products in the Binggou watershed, locations of snow stakes set during the Binggou watershed experiment, and the brightness temperature difference between the 18 GHz and 36 GHz from AMSR-E.**