# Peer review of "Evaluation of snow cover and snow depth on the Qinghai-Tibetan Plateau derived from passive microwave remote sensing"

_The Cryosphere, 2016_

## Referee Comment (RC1) · Anonymous Referee #1 · 18 Jan 2017

The Dai et al. paper used the MODIS fractional snow cover product, in situ observations, and airborne observation data to evaluate the accuracy of snow cover and snow depth derived from passive microwave remote sensing data and to analyze the possible causes of uncertainties, such as cold desert, soil temperature, atmospheric correction, spatial resolution and topography and snow characteristics. The analyses are well-organized, the results are quite specific, which can provide a reference for the use of passive microwave remote sensing retrieving snow cover and snow depth, especially in some complex land surface region, such as Tibetan Plateau. Despite of its significance, several issues still need to be resolved before a publication to The Cryosphere. The paper needs thorough English edits and I only catch a few below,

[Figure]

and some qualitative discussions are also needed.

Below are some general comments:

Introduction:

The first paragraph needs to rewrite. 'The Qinghai-Tibetan plateau (QTP) is considered the third pole of the world; its snow cover is an indicator of global climate change (Kang et al., 2010; Wu et al., 2003; Wang et al., 2015), and snow cover variation impacts the near ground air temperature and precipitation in Eurasia and across the Northern Hemisphere (Zhang et al, 2004; Lüet al., 2008; You et al.,2011).'This part should be more focused on the topic of study and does not need to include everything that does not link to the topic of snow cover change in QTP.

P2L17 remove should be replaced by eliminate.

P3L1-4 the part is makes no sense for the topic of this paper, should be removed.

P3L24 mainly should be replaced by primarily.

2. Data

2.1 MODIS snow cover fraction

Snow cover fraction (SCF)——Fractional snow cover (FSC)

Which products version was used in your manuscript?

2.3 Meteorological station observations of snow depth

How the snow water equivalents were measured by meteorological station?

P6L19 18 and 36GHz, please keep the consistency of expression. Use K and Ka band or 18 and 36GHz all through the paper.

2.4 Field experiments

Three observation routes were presented in figure 1, but in the following comparisons,

you didn't compare the results of the observations in red and green line, why?

P7L18 Why was the total portion of all possibilities not 1? Please check the data.

When compared with MODIS snow cover fraction in figure 3, what is the relationship between the left and right figures? Please describe them clearly.

3. Evaluation methods and results

Your results involved AMSR2, but nothing results about AMSR2 appeared in 3.2.1.

P7L18 "12.2

P8L13 snow depth is low—snow depth is shallow.

P8L18 . . .with shallow snow—because of shallow snow.

P9L2-3 'Snow cover conditions were derived from AMSR-E or AMSR2 at grids that contained meteorological stations and were compared with observations.' Please rewrite.

P9L4 'points'. I think here 'samples' should be better.

Table 1 2. Please use confusion matrix to introduce how the accuracy values calculate.

Please add the trend line and correlation coefficient in fig. 6a.

3.2.2 Comparison with field observations

When the estimated snow depth was compared with the observation along the observation route, do you consider the snow cover fraction as in Binggou watershed?

P10L4 'Compared with these snow depth.'Please replace the snow depth to samples.

P10L6 'deep'—'thick or heavy'

P10L18 'Snow depth in a PM grid is reflected in the dense sample and snow cover fraction across the QTP.' What is your meaning?

4. Sources of error

4.1 Cold desert

How do you know where is the cold desertïij§what is your base?

P11L1 'Take the Tuotuohe station for example'—'Take the Tuotuohe station as a example'.

What are the locations of Nyalam station (Id: 55655) in Section 3.2.1, Batang station (Id: 56247) in section 4.2, Tuotuohe station (Id:56004) in section 4.1, please label them on the fig. 1.

Fig. 8a. You mentioned the'significant', you have to do the F test, and also add the test value to fig. 8a.

P11L14'Therefore, the ground temperature is the main reason cause the change in TBD'.

P11L25 ...is key to improving the...—is key to improve the...

5 Conclusions

You need separate your conclusions into two parts, a discussion section and conclusions section, to put your results in the big picture of literature, how your results differ from, similar as, or extent in certain degree of the current literature. You also need to include a paragraph on the possible explanations to the observed change, difference, or extension.

Please also note the supplement to this comment:
http://www.the-cryosphere-discuss.net/tc-2016-262/tc-2016-262-RC1-supplement.pdf

---

## Author Comment (AC1) · 8 Feb 2017

Dear reviewer and editor,

Thanks for your positive comments and very useful recommendations to improve our manuscript. We have carefully modified the manuscript based on your suggestions and made response one by one. The main modifications consist of

1 Section 5 discussion was added putting our results in the big picture of literature.

2 Fig 1 and Fig 6 were revised according to your suggestions.

3 The confusion matrix of table 1 and table 2 were added.

4 Fig 3 was further clarified.

The Dai et al paper used the MODIS fractional snow cover product, in situ observations, and airborne observation data to evaluate the accuracy of snow cover and snow depth derived from passive microwave remote sensing data and to analyze the possible causes of uncertainties, such as cold desert, soil temperature, atmospheric correction, spatial resolution and topography and snow characteristics. The analyses are well-organized, the results are quite specific, which can provide a reference for the use of passive microwave remote sensing retrieving snow cover and snow depth, especially in some complex land surface region, such as Tibetan Plateau. Despite of its significance, several issues still need to be resolved before a publication to The Cryosphere. The paper needs thorough English edits and I only catch a few below, and some qualitative discussions are also needed.

Thank you very much for your reviewing our manuscript. We appreciated it that you gave a positive comments and very useful suggestions to improve our manuscript. We made a modification according to your suggestions and comments. The point-by-point revisions are as follows:

**Below are some general comments:**

**Introduction:**

The first paragraph needs to rewrite. 'The Qinghai-Tibetan plateau (QTP) is considered the third pole of the world; its snow cover is an indicator of global climate change (Kang et al., 2010; Wu et al., 2003; Wang et al., 2015), and snow cover variation impacts the near ground air temperature and precipitation in Eurasia and across the Northern Hemisphere (Zhang et al, 2004; Lüet al., 2008; You et al.,2011).' This part should be more focused on the topic of study

and does not need to include everything that does not link to the topic of snow cover change in QTP.

RE:The Qinghai-Tibetan plateau (QTP) is considered the third pole of the world; its snow cover is an indicator of global climate change (Kang et al., 2010; Wu et al., 2003; Wang et al., 2015), and snow cover variation impacts the near ground air temperature and precipitation in Eurasia and across the Northern Hemisphere (Zhang et al, 2004; Lüet al., 2008; You et al.,2011).' was revised to

"The Qinghai-Tibetan plateau (QTP) is considered the third pole of the world and the Asian water tower (Kang et al., 2010; Wu et al., 2003; Wang et al., 2015; Immerzeel et al., 2010; Xu et al., 2008). Snow cover over it plays a significant role in the climate change and hydrological circle."

P2L17 remove should be replaced by eliminate.

RE:It was revised.

P3L1-4 the part is makes no sense for the topic of this paper, should be removed.

RE:This part was removed according to your suggestion.

P3L24 mainly should be replaced by primarily.

RE:It was revised.

**2. Data**

2.1 MODIS snow cover fraction

Snow cover fraction (SCF)-----Fractional snow cover (FSC)

Which products version was used in your manuscript?

RE:Fractional snow cover is the name of the product, but snow cover fraction is better to describe the content. In the text, MOD10A1 and MYD10A are of fractional snow cover products.

"The Terra/Aqua MODIS Level 3, 500m daily snow cover products (MOD10A1 and MYD10A1) were obtained from the National Snow and Ice Data Center (NSIDC) from 1 January, 2003 to 31 December, 2014 (Hall et al., 2006). The snow cover fraction (SCF) product derived from MODIS is generated" was revised to "The Terra/Aqua MODIS Level 3, 500m daily fractional snow cover products (MOD10A1 and MYD10A1) were obtained from

the National Snow and Ice Data Center (NSIDC) from 1 January, 2003 to 31 December, 2014 (Hall et al., 2006). These products derived from MODIS were generated".

**2.3 Meteorological station observations of snow depth**

How the snow water equivalents were measured by meteorological station?

RE:Thanks for your remind.

" On the fifth, tenth, fifteenth, twentieth, twenty-fifth and the last day of a month (nearly every five days), snow water equivalents were measured using snow tube with a cross-sectional area of 100 $cm^2$ at the same time of snow depth measurement. And the record is also the mean value of three individual measurements. "  was added to the end of this paragraph.

P6L19 18 and 36GHz, please keep the consistency of expression. Use K and Ka band or 18 and 36GHz all through the paper.

RE:Thanks for your suggestion. 18 and 36 GHz were used all through the paper.

**2.4 Field experiments**

Three observation routes were presented in figure 1, but in the following comparisons, you didn't compare the results of the observations in red and green line, why?

RE:There was rare snow during these two investigation periods, passive microwave remote sensing seldom underestimate the snow cover over QTP.

Please refer to the beginning of section 3.2.2: "Observations from December of 2013 and May of 2014 indicated sparse snow along the observation route, a result also shown by AMSR2."

P7L18 Why was the total portion of all possibilities not 1? Please check the data.

RE:The total portion of all possibilities did equal to 1. Maybe the expression is not so clearly.

"grids with TBD more than 20 K showed 4.9 % snow-free area, 82.9 % snow area, and 12.2 % uncertainty area, including 6.1 % high possibility of snow cover area and 6.1 % high possibility of snow-free area." was revised to

"grids with TBD more than 20 K showed 4.9 % snow-free area, 82.9 % snow area, and 12.2 % uncertainty area. The uncertainty areas included 6.1 % high possibility of snow cover area and 6.1 % high possibility of snow-free area."

When compared with MODIS snow cover fraction in figure 3, what is the relationship between the left and right figures? Please describe them clearly.

RE:The probability in figure 3 left was the ratio of frequency of a certain group of SCF and the frequency of all SCF with cloud fraction less than 10 %.

1: Left figures described the spatial distribution of probabilities of SCF >10 % when TBDs were more than 20 K (figure 3 a), between 15 and 20 K (figure 3 b), between 10 and 15 K (figure 3 c), between 5 and 10 K (figure 3 d), and less than 5 K (figure 3 e). Right figures were the statistic results of different probabilities for all group of SCF all over the QTP. The first group with horizontal axis labeled by "> 10%" is the statistic result of the right figure. The red bar means the number of pixels with probability of "SCF >10%" between 0 and 0.1 all over the QTP, the yellow bar for between 0.1 and 0.5, the light blue bar for between 0.5 and 0.8, and the dark blue bar for more than 0.8. The other groups which labeled by ">30%", ">50%", ">70%", and ">90%" presented the same meaning corresponding their SCF range as "SCF >10%", but their spatial distribution were not presented in figures.

2: "The frequencies of each SCF with cloud fraction less than 10 % for each TBD group from 2003 to 2007 were computed, which we called TBD-SCF table. Based on the TBD-SCF table, the probability was calculated. The results are described in the TBD-SCF table, which is the basis for determining the likelihood of snow cover for each AMSR-E grid given a TBD. The flowchart for building the TBD-SCF table is provided in Fig. 2." was revised to

"The frequencies of each SCF with cloud fraction less than 10 % for each TBD group from 2003 to 2007 were computed, which we called TBD-SCF table. Based on the TBD-SCF table, the probability was calculated, which was the ratio of frequency of a certain group of SCF and the frequency of all SCF with cloud fraction less than 10 %. The flowchart for building the TBD-SCF table is provided in Fig. 2."

3: "The frequency histograms of SCF > 10 %, 30 %, 50 %, 70 %, and 90 % were calculated according to the TBD-SCF table (Fig. 3), and the spatial distribution of the frequency of SCF > 10 % corresponding to each TBD group is presented in Fig. 3." was revised to

"The probabilities of all SCF groups to all TBD groups were depicted in figure 3. Left figures

described the spatial distribution of probabilities of SCF >10 % when TBDs were more than 20 K (figure 3 a), between 15 and 20 K (figure 3 b), between 10 and 15 K (figure 3 c), between 5 and 10 K (figure 3 d), and less than 5 K (figure 3 e). Right figures were the statistic results of different probabilities for all group of SCF all over the QTP. The first groups with horizontal axis labeled by "> 10%" were the statistic result of the right figures. The red bar means the number of pixels with probability of "SCF >10%" between 0 and 0.1 all over the QTP, the yellow bar for between 0.1 and 0.5, the light blue bar for between 0.5 and 0.8, and the dark blue bar for more than 0.8. The other groups which labeled by ">30%", ">50%", ">70%", and ">90%" presented the same meaning corresponding their SCF range as "SCF >10%", but their spatial distribution were not presented in figures."

**3. Evaluation methods and results**

Your results involved AMSR2, but nothing results about AMSR2 appeared in 3.2.1.

RE:AMSR-E stopped in 2011, and AMSR2 started in 2012. The AMSR2 brightness temperatures from November, 2013 to March, 2014 were used to derive the snow depth in the field experiment areas, and the derived snow depths were compared with the field observations.

P7L18 "12.2 % uncertainty area, including 6.1 % high possibility of snow cover area and 6.1 % high possibility of snowfree area." Very confused.

RE:The total portion of all possibilities did equal to 1. Maybe the expression is not so clearly.

"grids with TBD more than 20 K showed 4.9 % snow-free area, 82.9 % snow area, and 12.2 % uncertainty area, including 6.1 % high possibility of snow cover area and 6.1 % high possibility of snow-free area." was revised to

"grids with TBD more than 20 K showed 4.9 % snow-free area, 82.9 % snow area, and 12.2 % uncertainty area. The uncertainty included 6.1 % high possibility of snow cover area and 6.1 % high possibility of snow-free area."

P8L13 snow depth is low---snow depth is shallow.

RE:It was revised.

P8L18 …with shallow snow---because of shallow snow.

RE:"Areas with shallow snow" means that in these areas snow is shallow. If it is changed to

because of, the meaning is also changed

P9L2-3 'Snow cover conditions were derived from AMSR-E or AMSR2 at grids that contained meteorological stations and were compared with observations.' Please rewrite.

RE:This sentence was revised to "Snow depths were derived from AMSR-E at grids that meteorological stations located on, and then were compared with station records."

P9L4 'points'. I think here 'samples' should be better.

RE:"points" is changed to "grids"

Table 1 & 2. Please use confusion matrix to introduce how the accuracy values calculate.

RE:Thanks for your suggestion. We added the confusion matrix in the paper. In the text, we had explained how to calculate the overestimate, underestimate, omission and commission. Following is the confusion matrix of the comparison of table 1 and table 2:

Table 1 confusion matrix

| Confusion matrix | Snow(MODIS) | No snow(MODIS) | Snow(station) | No snow(station) |
|---|---|---|---|---|
| Snow(PM) | 1367354 | 1749417 | 5139 | 27543 |
| No snow(PM) | 1232973 | 4597783 | 3656 | 144368 |

Table 2 confusion matrix

| Confusion matrix | Original | | After correction | |
|---|---|---|---|---|
| | Snow(MODIS) | No snow(MODIS) | Snow(MODIS) | No snow(MODIS) |
| Snow(PM) | 1367354 | 1749417 | 1023344 | 901632 |
| No snow(PM) | 1232973 | 4597783 | 1586860 | 5441964 |

Please add the trend line and correlation coefficient in fig. 6a.

RE:Thanks for your suggestion. Fig. 6a was changed to

"The results showed that AMSR-E overestimates snow depths across the QTP" was revised to "The results showed that the correlation coefficient between them was 0.124, and AMSR-E overestimates snow depths across the QTP"

3.2.2 Comparison with field observations

When the estimated snow depth was compared with the observation along the observation route, do you consider the snow cover fraction as in Binggou watershed?

RE:At the first paragraph of this section, we added "According to MODIS fractional snow cover products, snow cover fraction of the pixels that these points located in ranged from 0.5 to 0.9 when the observations showed snow. If the snow cover fraction is considered in the comparison, the bias is 1.77cm, RMSE is 5.66 cm."

P10L4 'Compared with these snow depth.' Please replace the snow depth to samples.

RE:It was revised.

P10L6 'deep' --- 'thick or heavy'

RE:It was revised to thick.

P10L18 'Snow depth in a PM grid is reflected in the dense sample and snow cover fraction across the QTP.' What is your meaning?

RE:This sentence was revised to "Evaluation of PM snow depth over QTP requires dense sampling in a whole pixel."

**4. Sources of error**

**4.1 Cold desert**

How do you know where is the cold desert?what is your base?

RE:The cold desert can be identified by polarization difference. When the polarization difference is more than 18K, it can be regarded as cold desert, based on Grody and Basist (1996).

"Based on the classification criterion of Grody and Basist (1996), cold desert presented large polarization. There are large areas of cold desert on the middle and Northwest part of QTP, which also showed scattering" was added to the start of section 4.1.

"Take the Tuotuohe station (Id:56004) for example; this station is located in a desert area, during the winter, sand scatters the microwave signal and presents weak scattering features." was revised to "Take the Tuotuohe station (Id:56004, Latitude: 34.22N, Longitude: 92.43E,

Fig. 1) which located on these areas (Fig. 1), for example; during the winter, ground scatters the microwave signal and presents weak scattering features."

P11L1 'Take the Tuotuohe station for example' --- 'Take the Tuotuohe station as a example'. What are the locations of Nyalam station (Id: 55655) in Section 3.2.1, Batang station (Id: 56247) in section 4.2, Tuotuohe station (Id:56004) in section 4.1, please label them on the fig. 1.

RE:Thanks for your remind. We added the longitude and latitude of these two stations, and presented the location in figure 1.

[Figure]

Fig. 8a. You mentioned the 'significant', you have to do the F test, and also add the test value to fig. 8a.

RE:Thanks for your suggestion. The $R^2$ showed there was a strong relationship between TB36V and TBD. According to your suggestion, we also did F test, and found their relationship presented significant correlation at the confidence level of 0.95.

In the first paragraph of section 4.2, "TB36V is sensitive to topsoil temperature (Holmes et al., 2009; Zeng et al., 2015). Statistical analysis between TBD (K) and TB36V at 109 stations showed that TBD has a significant negative correlation with TB36V (Fig. 8 a)" was revised to "TB36V is sensitive to topsoil temperature (Holmes et al., 2009; Zeng et al., 2015). Statistical analysis between TBD (K) and TB36V at 109 stations showed that TBD has a significant negative correlation with TB36V (Fig. 8 a) at the confidence level of 0.95"

P11L14 'Therefore, the ground temperature is the main reason cause the change in TBD'.

RE:"The ground temperature is the main contributor to the increase in TBD." was revised to "the ground temperature is also a main reason causing large TBD."

P11L25 ...is key to improving the...---is key to improve the...

RE:It was corrected.

**5 Conclusions**

You need separate your conclusions into two parts, a discussion section and conclusions section, to put your results in the big picture of literature, how your results differ from, similar as, or extent in certain degree of the current literature. You also need to include a paragraph on the possible explanations to the observed change, difference, or extension.

RE:Thanks for your suggestions. We added section 5 discussions, and made some modification in section 6 conclusion. In the section 1 and section 4, we also made some revisions to make analysis clearer.

[revised manuscript text omitted]

6:" Furthermore, frozen soil is also a scatter; the TBD is influenced by frozen soil, and increases with the increase of ice content. Based on experiments, the TBD of frozen soil with high ice content could reach up to over 10 K (Jin et al., 2017)" were added in the section 4.2.

7: The added the references

"Derksen, C., Walker, A. and Goodison, B.: Evaluation of passive microwave snow water

equivalent retrievals across the boreal forest/tundra transition of western Canada. Remote Sens Environ, 96(3-4): 315-327, 2005.

Smith, T. and Bookhagen, B.: Assessing uncertainty and sensor biases in passive microwave data across High Mountain Asia. Remote Sensing of Environment, 181: 174-185, 2016.

Armstrong, R.L., Brodzik, M.J.: Hemispheric-scale comparison and evaluation of passive-microwave snow algorithms. Ann Glaciol., 34: 38-44, 2002.

Vander Jagt, B.J., Durand, M.T., Margulis, S.A., Kim, E.J., Molotch, N.P.: On the characterization of vegetation transmissivity using LAI for application in passive microwave remote sensing of snowpack. Remote Sens Environ, 156: 310-321, 2015.

Pulliainen, J.T., Grandell, J., Hallikainen, M.T.: HUT snow emission model and its applicability to snow water equivalent retrieval. Ieee T Geosci Remote, 37(3): 1378-1390, 1999.

Langlois, A., A. Royer, F. Dupont, A. Roy, Goita K. and Picard G.: Improved Corrections of Forest Effects on Passive Microwave Satellite Remote Sensing of Snow Over Boreal and Subarctic Regions. Ieee T Geosci Remote, 49(10): 3824-3837, 2011.

Foster, J. L., Chang A. T. C. and Hall D. K.: Comparison of snow mass estimates from prototype passive microwave snow algorithm, a revised algorithm and a snow depth climatology. Remote Sens Environ, 62(2): 132-142, 1997.

Goita, K., Walker A. E.  and Goodison B. E.: Algorithm development for the estimation of snow water equivalent in the boreal forest using passive microwave data. Int J Remote Sens,, 24(5): 1097-1102, 2003.

England, A.W.: Relative influence upon microwave emissivity of fine-scale stratigraphy, internal scattering, and dielectric properties. Pure Appl Geophys, 114(2): 287-299, 1976"
were also added to the end of references.

.

---

## Referee Comment (RC2) · Anonymous Referee #2 · 13 Feb 2017

Snow cover is one of most important factors affected the land-atmosphere interaction and water budget on the Qinghai-Tibetan Plateau. And snow depth estimated from passive microwave remote sensing has been reported with uncertainties and the source of them has been discussed for many years. This paper utilized multi-source data including MODIS snow cover data, station measurements and snow course data, to evaluate the snow depth derived from AMSR-E and AMSR2 based on a spectral gradient algorithm. Many factors have impacts on the discrimination of snow from snow-free ground, and the linear relationship between the brightness temperature difference and snow depth. This is an interesting work. The impacts factors affected snow depth retrieval algorithm have been discussed thoroughly in this work. But the

authors still need to consider the following questions.

General comments/suggestions: 1. The threshold of MODIS snow fraction >10% is too small to definite snow covered surface. I suggest snow fraction should use a larger value. Since passive microwave remote sensing cannot detect snow when the grid is covered with 10% snow fraction. 2. My another question is on the explanation of soil temperature effect on snow depth algorithm. The authors should justify the explanation on this. The reason might be difficultly discriminate dry snow from frozen soil. The soil temperature could be contributed to discriminate dry snow from frozen soil. There are similar scattering existing between dry snow and frozen soil. While the soil temperature is different between them as the authors stated in this work. The soil temperature might be higher when the surface is covered with snow. 3. Section 2.1, "The relationship equation is SCF = 0.06 + 1.21 * NDSI". This equation is confusing given that for MOD10A1 in C5 and C6 and MYD10A1 in C6, SCF =-0.01 + (1.45 * NDSI6) and for MYD10A1 in C5, SCF =-0.64 + (1.91 * NDSI7). Please check it again and reference it to published papers. In which collection the NASA MODIS standard snow cover products were collected should be clearly stated as well because of some differences among different collections.

References: V. V. Salomonson and I. Appel, "Development of the Aqua MODIS NDSI fractional snow cover algorithm and validation results," in IEEE Transactions on Geoscience and Remote Sensing, vol. 44, no. 7, pp. 1747-1756, July 2006. G. A. Riggs and D. K. Hall, "MODIS Snow Products Collection 6 User Guide," 2015.

4. Details of the snow identification algorithm is suggested to be more specific, such as the criteria of dry snow determination, how the "offset" value was determined in the equation "snow depth (cm) = 0.7*(TB18H-TB36H-5)+offset". The thresholding snow identification algorithm used in this paper was referenced to Che et al.(2008), in which threshold values were separately determined using SMMR and SMM/I data. In addition, AMSR-E and AMSR2 employed a same suite of thresholds in this paper. Given different design characteristics and other factors causing bias among sensors, it should

be explained why this threshold-based method was applied consistently across sensors before inter-calibration. 5. As far as I know, both commission error and over-estimation error in snow mapping are related with false snow detection on snow-free ground. Equations are needed to clarify how commission, omission, overestimation and underestimation errors were calculated in this paper. 6. Was wet snow excluded from ground observations before the evaluation of AMSR-E/AMSR2 snow depth? The capability of detecting snow depth from passive microwave data is limited by liquid-water content in snow cover. Also, it is better to discuss the effect of forest cover on evaluation of snow depth retrieval algorithm.

Minor comments/suggestions: 1. I suggested that it would be better that "Passive microwave (PM)" used in this manuscript replaced with "PMW". Since "PM" also is the abbreviation of Post Meridiem, meaning after midday. 2. Page5 Line 25, there is a typo appeared on "Cold desert: TB19V-TB18V >=18 (K) . . .".

3. In Fig. 1., please check if several selected meteorological stations on top left are out of the range of the Tibetan Plateau. 4. As for the title of Fig. 4 and Fig. 8, there are some sub-figures in the sequence of alphabets. But these alphabets are supposed to be ahead of sub-titles. 5. In Fig. 7, the value of X-axis is missed. 6. In Fig. 8, it's better use the density plot i.e. scatter points in different color varying with point density, to see whether most points are gathering around the 1:1 line. 7. In Fig. 9., the geo-location of Binggou watershed would be more understandable with frame ticks and latitude/longitude labels.

---

## Author Comment (AC2) · 22 Feb 2017

Dear reviewer and editor,

Thanks for your comments and very useful recommendations to improve our manuscript. We have carefully revised them based on your suggestions and made responses one by one.

Snow cover is one of most important factors affected the land-atmosphere interaction and water budget on the Qinghai-Tibetan Plateau. And snow depth estimated from passive microwave remote sensing has been reported with uncertainties and the source of them has been discussed for many years. This paper utilized multi-source data including MODIS snow cover data, station measurements and snow course data, to evaluate the snow depth derived from AMSR-E and AMSR2 based on a spectral gradient algorithm. Many factors have impacts on the discrimination of snow from snow-free ground, and the linear relationship between the brightness temperature difference and snow depth. This is an interesting work. The impacts factors affected snow depth retrieval algorithm have been discussed thoroughly in this work. But the authors still need to consider the following questions. General comments/suggestions:

1. The threshold of MODIS snow fraction >10% is too small to definite snow covered surface. I suggest snow fraction should use a larger value. Since passive microwave remote sensing cannot detect snow when the grid is covered with 10% snow fraction.

Re: Thanks for your suggestion. We calculated the overestimation, underestimation, commission and omission when the MODIS fractional snow cover more than 30% and 50% was defined to be snow cover, and the descriptions were also added in the section 3.1.

" If pixels with SCF >0.3 is assumed to be snow pixel, the overestimation, underestimation, commission and omission errors are 72.2%, 0.9%, 17.5% and 9.9%, respectively. If SCF > 0.5 is the threshold between snow and no snow, they are 83.2%, 0.4 %, 19.5% and 7.2%, respectively.    The overestimation and commission errors increase with the increase of the threshold, and the underestimation and omission errors decrease. " was added in the section 3.1 P8 L5-9.

And we explained the Fig.3 which contained the information of different SCF thresholds:

"The frequency histograms of SCF > 10 %, 30 %, 50 %, 70 %, and 90 % were calculated according to the TBD-SCF table (Fig. 3), and the spatial distribution of the frequency of SCF > 10 % corresponding to each TBD group is presented in Fig. 3." was revised to

 "The probabilities of all SCF groups to all TBD groups were depicted in figure 3. Left figures described the spatial distribution of probabilities of SCF >10 % when TBDs were more than 20 K (Fig.3 a), between 15 and 20 K (Fig.3 b), between 10 and 15 K (Fig. 3 c), between 5 and 10 K (Fig.3 d), and less than 5 K (Fig. 3 e). Right figures were the statistic results of different probabilities for all group of SCF all over the QTP. The first groups with horizontal axis labeled by "> 10%" were the statistic result of the right figures. The red bar means the number of pixels with probability of "SCF >10%" between 0 and 0.1 all over the QTP, the yellow bar for between 0.1 and 0.5, the light blue bar for between 0.5 and 0.8, and the dark blue bar for more than 0.8. The other groups which labeled by ">30%", ">50%", ">70%", and ">90%" presented the same meaning corresponding their SCF range as "SCF >10%", but their spatial distribution were not presented in figures."P8 L9-16

2. My another question is on the explanation of soil temperature effect on snow depth algorithm. The authors should justify the explanation on this. The reason might be difficultly discriminate dry snow from frozen soil. The soil temperature could be contributed to discriminate dry snow from frozen soil. There are similar scattering existing between dry snow and frozen soil. While the soil temperature is different between them as the authors stated in this work. The soil temperature might be higher when the surface is covered with snow.

Re: Thanks for your remind. We added the explanation in the section 5 Discussions:

"This study also showed that the TBD was mainly controlled by soil temperature. It has a strong negative correlation with brightness temperature at 36 GHz for vertical polarization which is the most sensitive to ground surface temperature.

Brightness temperature at 36 GHz is much more sensitive to the land surface temperature than 18 GHz. With the surface temperature declines, the brightness temperature at 36 GHz decreases quickly. But the brightness temperature at 18 GHz keeps stable, because it is influence by temperature at deeper layer of soil. Therefore, brightness temperature at 36GHz is lower than that of 18GHz, and the difference between them increases with the decrease of temperature of surface soil. Moreover, because of the freeze/thaw cycle of surface soil, the frozen soil becomes incompact and dry. The fine-scale soil and sand particles are scatters which also weaken the brightness temperature at 36GHz (England, 1976). In the northwest of the QTP, the surface temperature is very low and the polarization difference larger than 30 K which is the characteristics of desert. Therefore, we inferred that the combined action of frozen soil and desert resulted in large TBD, and then caused the serious overestimation." Was added in the section 5.

3. Section 2.1, "The relationship equation is SCF = 0.06 + 1.21 * NDSI". This equation is confusing given that for MOD10A1 in C5 and C6 and MYD10A1 in C6, SCF =-0.01 + (1.45 * NDSI6) and for MYD10A1 in C5, SCF =-0.64 + (1.91 * NDSI7). Please check it again and reference it to published papers. In which collection the NASA MODIS standard snow cover products were collected should be clearly stated as well because of some differences among different collections.

References: V. V. Salomonson and I. Appel, "Development of the Aqua MODIS NDSI fractional snow cover algorithm and validation results," in IEEE Transactions on Geoscience and Remote Sensing, vol. 44, no. 7, pp. 1747-1756, July 2006.

G. A. Riggs and D. K. Hall, "MODIS Snow Products Collection 6 User Guide," 2015.

Re: Thanks for your correction. We used the collection 5 data, and modified the description based on Salomonson and Appel (2006).

"The relationship equation is SCF = 0.06 + 1.21 * NDSI7, and it was developed over three different snow covered regions" was revised to

"The relationship equation is SCF = a+ b* NDSI, and the coefficients "a" and "b" and NDSI vary with sensors. Coefficients "a" and "b" are -0.01 and 1.45 for MODA1, and -0.64 and 1.91 for MYDA1, respectively. NDSI is the function of band 4 and band 6 for Terra (MODA1), and of band 4 and band 7 for Aqua (MYDA1) (Salomonson and Appel, 2006)."

4. Details of the snow identification algorithm is suggested to be more specific, such as the criteria of dry snow determination, how the "offset" value was determined in the equation "snow depth (cm) = 0.7*(TB18H-TB36H-5)+offset". The thresholding snow identification algorithm used in this paper was referenced to Che et al.(2008), in which threshold values were separately determined using SMMR and SMM/I data. In addition, AMSR-E and AMSR2 employed a same suite of thresholds in this paper. Given different design characteristics and other factors causing bias among sensors, it should be explained why this threshold-based method was applied consistently across sensors before inter-calibration.

Re: Thanks for your suggestion.

1 "In order to avoid the influence of liquid water, the descending orbit data was used and when the brightness temperature at 36GHz for vertical polarization was more than 265 K, the snowpack was set as wet" was added in section 2 P5l23-24.

2 "where, offset was monthly data which was used to decrease the influence of snow characteristics growth. They are -4.18, -3.58, -0.29, 2.15, 3.31 and 3.8 for Oct, Nov, Dec, Jan, Feb, Mar and Apr, respectively" was added after these criterion.

3 "These criterions were developed based on SSM/I (Grody and Basist, 1996). Based on the inter-sensor comparison between SSM/I and AMSR-E in Dai and Che (2010), and between AMSR-E and AMSR2 in Du et al., (2014), the brightness temperature from these sensors are close to each other. Therefore, in this study, these criterions were also applied for AMSR-E and AMSR2" was added in the end of section 2.2.

We also added corresponding references

Du, J., Kimball, J., Shi, J., Jones, L., Wu, S., Sun, R., and Yang, H.: Inter-Calibration of Satellite Passive Microwave Land Observations from AMSR-E and AMSR2 Using Overlapping FY3B-MWRI Sensor Measurements. Remote Sens., 6(9): 8594-8616, 2014.

Dai, L.Y., Che, T. : Cross-platform calibration of SMMR, SSM/I and AMSR-E passive microwave brightness temperature. P Soc Photo-Opt Ins, 7841, 2010.

5. As far as I know, both commission error and overestimation error in snow mapping are related with false snow detection on snow-free ground. Equations are needed to clarify how commission, omission, overestimation and underestimation errors were calculated in this paper.

Re: Thanks for your suggestion.

Given the bellowing table,

|  | Estimated snow | Estimated no snow |
|---|---|---|
| Observed snow | a | b |
| Observed no snow | c | d |

the general accuracy, overestimation error, underestimation error, commission error and omission error are calculated based on the following equations:

General accuracy=(a+d)/(a+b+c+d)

Overestimation error=c/(a+c)

Underestimation error=b/(b+d)

Commission error= d/(c+d)

Omission error=b/(a+b)

We added a confusion matrix in the table 1 and table 2.

Table 1 Errors in derived snow cover from AMSR-E based on MODIS snow cover fraction and meteorological stations (a), and corresponding confusion matrix (b).

(a)

|  | Overall accuracy | Commission | Omission | Overestimation | Underestimation |
|---|---|---|---|---|---|
| MODIS snow cover fraction | 66.7 % | 27.6 % | 47.4 % | 56.1 % | 21.1 % |
| Meteorological stations | 82.7 % | 16.0 % | 41.6 % | 84.3 % | 2.5 % |

(b)

| Confusion matrix | Snow(MODIS) | No snow(MODIS) | Snow(station) | No snow(station) |
|---|---|---|---|---|
| Snow(PM) | 1367354 | 1749417 | 5139 | 27543 |
| No snow(PM) | 1232973 | 4597783 | 3656 | 144368 |

Table 2 Errors in snow cover derived from AMSR-E data and atmosphere corrected AMSR-E data over the QTP, based on MODIS snow cover fraction (a), and corresponding confusion matrix (b)

(a)

|  | Overall accuracy | Commission | Omission | Overestimation | Underestimation |
|---|---|---|---|---|---|
| Original | 66.7 % | 27.6 % | 47.4 % | 56.1 % | 21.1 % |
| After atmosphere correction | 72.2 % | 14.2 % | 60.8 % | 46.8 % | 22.6 % |

| | (b) | | | |
|---|---|---|---|---|
| Confusion matrix | Original | | After correction | |
| | Snow(MODIS) | No snow(MODIS) | Snow(MODIS) | No snow(MODIS) |
| Snow(PM) | 1367354 | 1749417 | 1023344 | 901632 |
| No snow(PM) | 1232973 | 4597783 | 1586860 | 5441964 |

6. Was wet snow excluded from ground observations before the evaluation of AMSR-E/AMSR2 snow depth? The capability of detecting snow depth from passive microwave data is limited by liquidwater content in snow cover. Also, it is better to discuss the effect of forest cover on evaluation of snow depth retrieval algorithm.

Re: During the observation of March 2014, the wetness of snowpack was observed only on March 23 and 24 using snowfolk. On March 23, the wetness of the observation pits was 1-2%. On March 24, it was 0.1-0.5%. In this study, the brightness temperature of cold overpass time was used to derive snow depth. Therefore, we regarded them as dry snow. On March 23, the snow could not be identified by PMW due to the fresh snow.

Forest is a general problem for PMW to derive snow depth. In the QTP, forest mainly distributes in the southeast areas. Based on your suggestion, we compared the forest cover fraction and the distribution of errors, and found that the omission errors distributed in the southeast areas (Fig. 4b) which is also the main forest areas. Therefore, in the section 5 Discussion, we added the description:

"Although satellite-based passive microwave brightness temperature data have been used to monitor global and regional snow depth since the 1980s, there were still some uncertainties on the snow depth retrieval algorithm in the QTP. Based on existing research on the evaluation of PM products, forest and grain size were the main causes resulting in the low accuracy of PM algorithm. In the forest regions, snow depth was usually underestimated by PM, and many methods had been developed to overcome it (Forest et al., 1997; Vander et al., 2015; Pullianen et al., 1999; Che et al., 2016). In the QTP, forest mainly distribute in the southeast region with rare snow, therefore, it is not the dominant factor in the QTP. However, in the Fig 4 b, the northeast region with omission errors was the main forest areas. In this area, snowfall is a rare event, and volume scattering signal is weak, plus the forest cover, it is difficult for PMW to detect snow."

Minor comments/suggestions:

1. I suggested that it would be better that "Passive microwave (PM)" used in this manuscript replaced with "PMW". Since "PM" also is the abbreviation of Post Meridiem, meaning after midday.

Re: Thanks, it was revised through the paper including Fig. 2.

2. Page5 Line 25, there is a typoappeared on "Cold desert: TB19V-TB18V >=18 (K) . . .".

Re: Thanks, it was corrected.

3. In Fig. 1., please check if several selected meteorological stations on top left are out of the range of the Tibetan Plateau.

Re: Thanks, it was corrected. We removed the two stations on the top left.

4. As for the title of Fig. 4 and Fig. 8, there are some sub-figures in the sequence of alphabets. But these alphabets are supposed to be ahead of sub-titles.

Re: Thanks, it was modified.

5. In Fig. 7, the value of X-axis is missed.

Re: Thanks, we added the x axis lable.

6. In Fig. 8, it's better use the density plot i.e. scatter points in different color varying with point

density, to see whether most points are gathering around the 1:1 line.

Re: Fig. 8 is used to present the relationship between TB36v and TBD (K). The x-axis is TB36v (K), and y-axis is TBD (K), so there is no 1:1 line. From this figure, we can see that there is obvious negative relationship between TBD(K) and TB36v (K). In order to clarify its significance, we did an F test, and found the relationship is significant at the confidence level of 0.95.

7. In Fig. 9.,the geo-location of Binggou watershed would be more understandable with frame ticks and latitude/longitude labels.

Re: Thanks, we added the longitude and latitude.